# Phase-engineered cathode for super-stable potassium storage

Lichen Wu[1,2], Hongwei Fu[1,2], Shu Li[1,2], Jian Zhu [3] ✉, Jiang Zhou [4], Apparao M. Rao [5], Limei Cha [6,7,8] ✉, Kunkun Guo[9], Shuangchun Wen[1] & Bingan Lu [1,2] ✉

The crystal phase structure of cathode material plays an important role in the cell performance. During cycling, the cathode material experiences immense stress due to phase transformation, resulting in capacity degradation. Here, we show phase-engineered $VO_2$ as an improved potassium-ion battery cathode; specifically, the amorphous $VO_2$ exhibits superior K storage ability, while the crystalline M phase $VO_2$ cannot even store $K^+$ ions stably. In contrast to other crystal phases, amorphous $VO_2$ exhibits alleviated volume variation and improved electrochemical performance, leading to a maximum capacity of 111 mAh g$^{-1}$ delivered at 20 mA g$^{-1}$ and over 8 months of operation with good coulombic efficiency at 100 mA g$^{-1}$. The capacity retention reaches 80% after 8500 cycles at 500 mA g$^{-1}$. This work illustrates the effectiveness and superiority of phase engineering and provides meaningful insights into material optimization for rechargeable batteries.

To improve the electrochemical performance of energy storage materials, various techniques have been explored from the perspective of composition, morphology, dimension, and size through common methods such as doping[1], coating[2], and heterostructure[3]. Notwithstanding this progress, the dispersion of atoms in the nanoparticle matrix ultimately dictates its properties. Consequently, phase engineering directly determines the properties and functioning of nanomaterials[4,5]. The unconventional phase materials often have unique properties[6], and the amorphous phase is one of them. Unlike the crystalline phase, the amorphous configuration provides an open architecture with abundant structural defects, inherent isotropy, and significant pseudocapacitance contributions. In addition to accommodating more uniform volume variation, the amorphous configuration also renders more exposed ion channels, facilitates rapid ion intercalation, and provides low internal energy and superior chemical stability[7,8]. A few studies have focused on amorphization for cation storage and explained the advantages of the amorphous phase from the perspective of theoretical calculations[9–13]. The superiority of using amorphous phases in energy storage applications can be further verified and solidified if more convincing experimental results can be combined with theoretical calculations in the context of continuing previous research trends.

Previous studies revealed that $VO_x$ could not be converted into V metal even when discharged to a very low voltage, tending to the intercalation reaction rather than the conversion reaction[14]. This feature considerably alleviates the volume expansion during cycling. In addition, $VO_x$ also has a high specific capacity, moderate operating potential, and low cost. From the frontier orbital theory, V has a strong electronic correlation, which provides a wide range of physical and chemical properties for vanadium composites. Particularly, vanadium possesses only three $d$-orbital electrons, which will not be affected by the Jahn–Teller effect like $Mn^{3+}$[15,16]. Interestingly, $VO_2$ exhibits rapid

[1]School of Physics and Electronics, Hunan University, Changsha 410082, PR China. [2]State Key Laboratory of Advanced Design and Manufacturing for Vehicle Body, Hunan University, Changsha 410082, PR China. [3]College of Chemistry and Chemical Engineering, Hunan University, Changsha 410082, PR China. [4]School of Materials Science and Engineering, Central South University, Changsha 410083, PR China. [5]Department of Physics and Astronomy, Clemson Nanomaterials Institute, Clemson University, Clemson, SC 29634, USA. [6]Materials Science and Engineering program, Guangdong Technion–Israel Institute of Technology, Shantou 515063, PR China. [7]Materials Science and Engineering program, Technion–Israel Institute of Technology, Haifa 32000, Israel. [8]MATEC key lab, Guangdong Technion–Israel Institute of Technology, Shantou 515063, PR China. [9]College of Materials Science and Engineering, Hunan University, Changsha 410082, PR China. ✉e-mail: jzhu@hnu.edu.cn; cha.limei@gtiit.edu.cn; luba2012@hnu.edu.cn

metal ion diffusion and superior rate capability than other vanadium oxides, making it an appropriate electrode material for lithium-ion batteries[17,18], sodium-ion batteries[19,20], and likewise for zinc-ion batteries[21,22]. It is worth noting that most of the research in refs. [17–22] focused on the M and B phases of VO₂. The M phase VO₂ crystallizes into the monoclinic $P2_1/c$ space group with a tunnel structure via corner-sharing VO₆ octahedra. The structure contains alternate shorter (0.265 nm) and longer (0.312 nm) length $V^{4+}$–$V^{4+}$ pairs along the $a$-axis[23]. Moreover, the B phase VO₂ belongs to the triclinic crystal structure with space group $C2/m$, in which VO₆ octahedra edge-sharing bilayers structure contains one-dimensional tunnels allowing fast ion diffusion[24].

In this study, we pick VO₂ as an example cathode for potassium-ion batteries (PIBs) with material modification from the phase engineering perspective. We utilize carbon fiber cloth (CF) as a substrate to further enhance electronic conductivity. Compared to the crystalline VO₂, amorphous VO₂ on carbon fiber cloth (VO₂ (a)) can achieve higher capacity (111 mAh g⁻¹ at 20 mA g⁻¹) and remarkable cycling performance (over 8 months at 100 mA g⁻¹ and 80% capacity retention after 8500 cycles at 500 mA g⁻¹). Furthermore, this study strongly integrates theoretical calculations and experiments to clarify that VO₂ (a) exhibits super-stable K storage capability, better electronic conductivity, and a lower ion diffusion barrier than its crystalline analogues. In sharp contrast, the B phase VO₂ has a poor capacity and cycle life, while the M phase VO₂ cannot even store K⁺ ions stable. This study demonstrates that a rational understanding and control of phase engineering is an

effective method to enhance material properties and provides an alternative strategy for developing rechargeable batteries.

## Results

### Potassium storage capability of VO₂ in different phases

The hydrothermal method and subsequent annealing processes prepared the M phase VO₂ on carbon fiber cloth (VO₂ (M)). The high-angle annular dark-field (HAADF) image in scanning transmission electron microscopy (STEM) with corresponding elemental mappings of VO₂ (M) at discharged state are illustrated in Fig. 1a–d. The elemental mappings indicate that the V and O elements are homogeneously distributed in the selected portion, while the K element is only found in some regions and is negligible in others. Figure 1e is the high-resolution transmission electron microscopy (HRTEM) image of the region with K (the tan box in Fig. 1a), and the corresponding fast Fourier transform (FFT) pattern in the inset shows that this area is amorphous. By contrast, the HRTEM of the region without K (the dark blue box in Fig. 1a) in Fig. 1f displays an ordered structure. In its corresponding FFT pattern, the (100), (−202), and (−102) planes of VO₂ (M) are indexed and marked by different rings in blue, yellow, and red, respectively. The simulated diffraction pattern (using the tool cellViewer by CrysTBox software[25]) along [010] zone axis of VO₂ (M) is shown in Fig. 1g, where both the symmetry of pattern and d-spacings in reciprocal space are consistent with the experimental FFT pattern. Figure 1h schematically illustrates both pristine VO₂ (a) (left) and K-embedded VO₂ (a) (right). The crystal structure of VO₂ (M) is

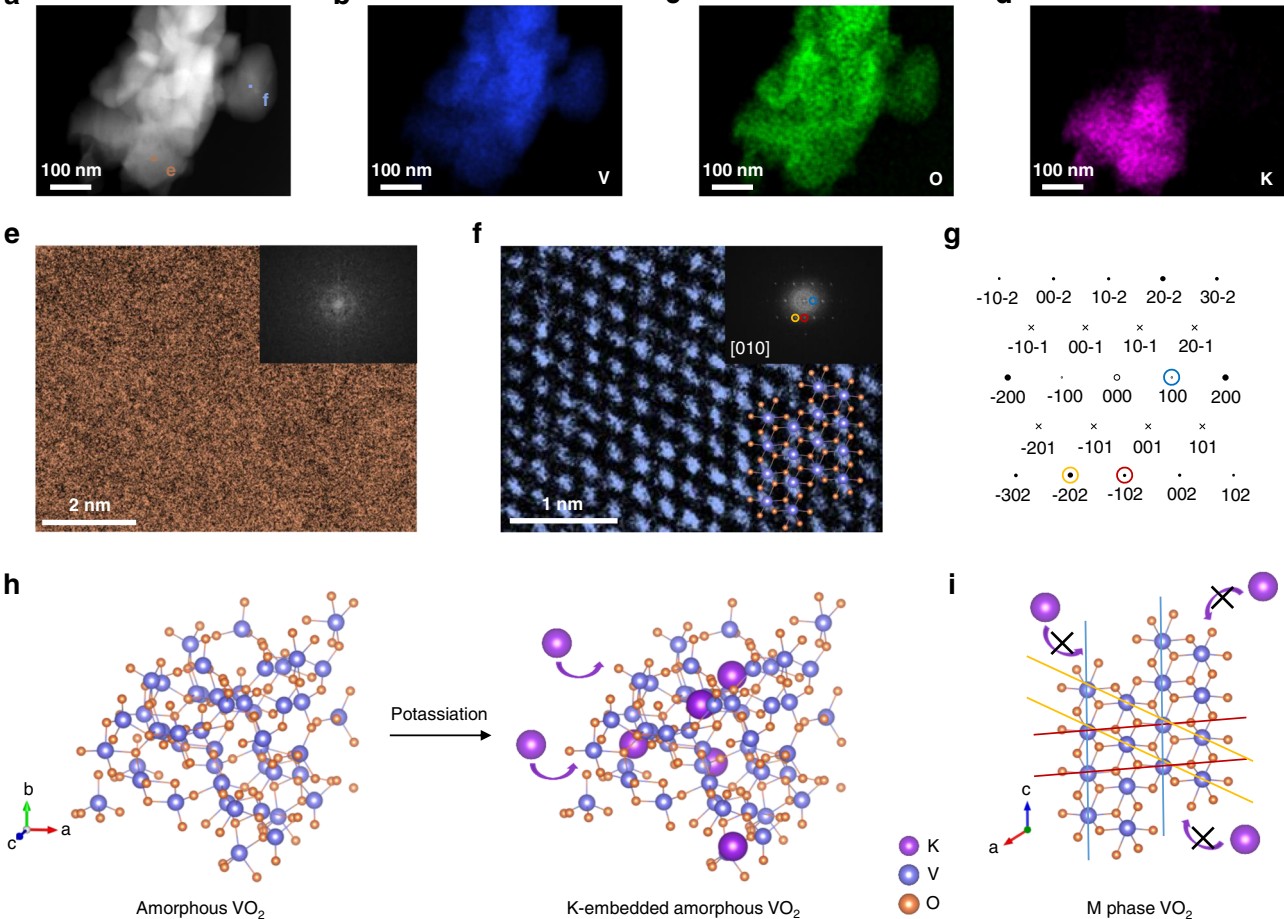

**Fig. 1 | Investigation of potassium storage capability of VO₂ in different phases.** **a** HAADF-STEM image of VO₂ (M) with corresponding elemental mapping of (**b**) V, (**c**) O, and (**d**) K. **e**, **f** The HRTEM images of the tan and dark blue squared areas in Fig. 1a. Inset: The corresponding FFT patterns. g) The simulated diffraction pattern of VO₂ (M) along the [010] zone axis. Schematic illustrations for (**h**) pristine VO₂ (**a**), K-embedded VO₂ (**a**), and (**i**) crystalline VO₂ (**M**).

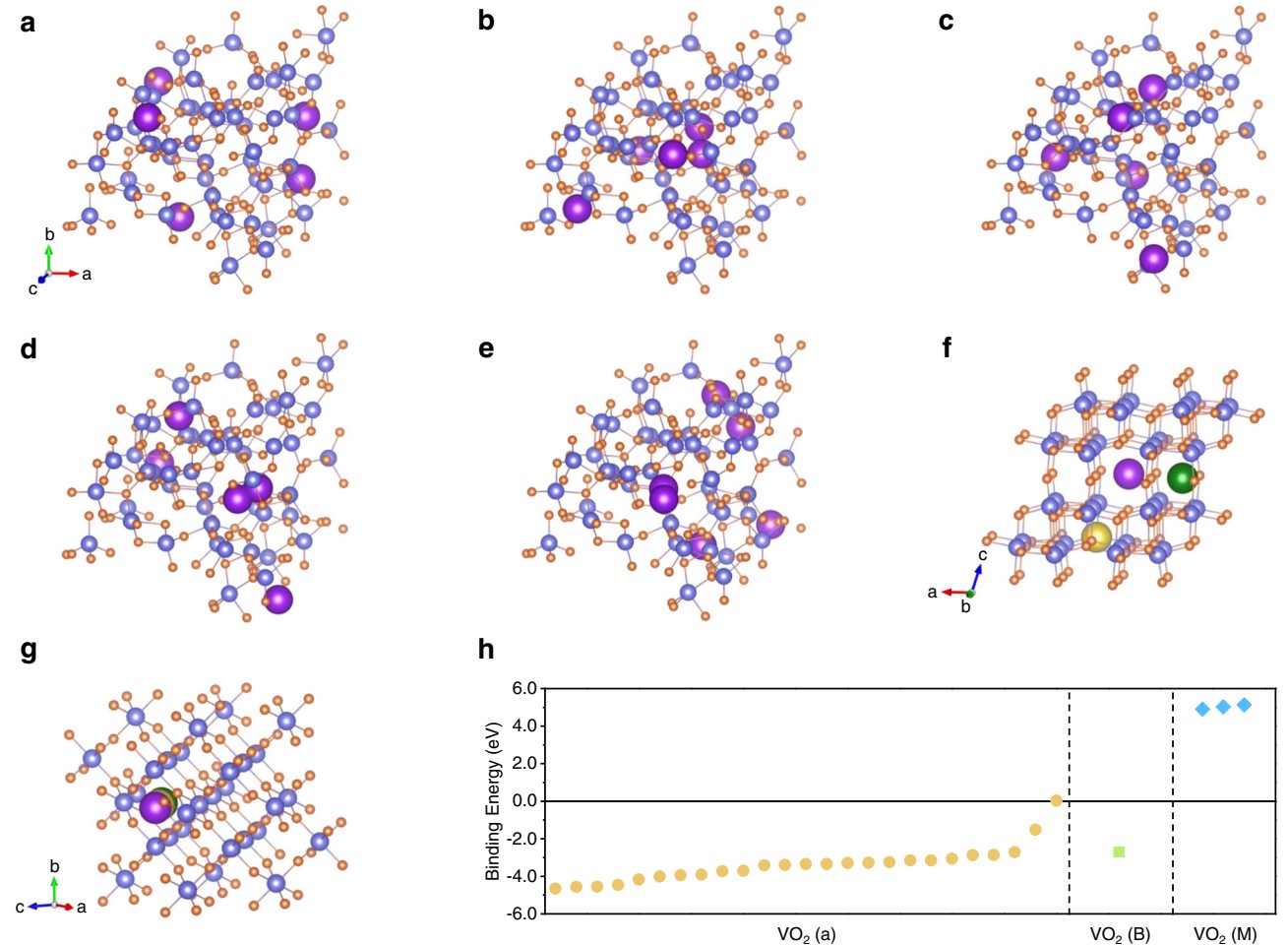

**Fig. 2 | Comparison of the binding energies of K⁺ ions for different VO₂ phases.** The K⁺ ions insertion sites in (**a**–**e**) VO₂ (a), (**f**) VO₂ (B), and (**g**) VO₂ (M). **h** The binding energies of K⁺ ions at different positions in VO₂ (a), VO₂ (B), and VO₂ (M).

displayed in Fig. 1i, in which K⁺ ions cannot enter the crystal lattice effectively during the discharge process. The different colored lines in Fig. 1i and the circled spots with the corresponding colors in FFT patterns indicate the same crystal planes of VO₂ (M).

From the above information, it can be presumed that the crystalline VO₂ (M) phase will exhibit no potassium storage capacity, while the VO₂ (a) phase could store potassium. The density functional theory (DFT) calculations were carried out to verify this presumption further. In addition to VO₂ (a) and VO₂ (M), we also examined a B phase VO₂ on carbon fiber cloth (VO₂ (B)), which is believed to be a promising candidate for high-performing electrode material. The hypothetical amorphous structure VO₂ was constructed from ab initio molecular dynamics (AIMD) simulation[26]. To identify potential insertion sites in VO₂ (a), 49 potassium storage sites were selected in the amorphous configuration according to the principle of uniform distribution, and their distances from V and O atoms were farther than 1.45 Å. After structural optimization, 25 sites with different positions remained. In Fig. 2a–e, the insertion sites of VO₂ (a) are shown in groups of five positions from low to high binding energy. Three non-equivalent insertion sites indicated by different colors are present for both VO₂ (B) and VO₂ (M) (Fig. 2f, g), which differ by coordination number and bond distances[27,28]. Multiple different insertion sites of K⁺ ions result in varying binding energies, as depicted in Fig. 2h. The negative binding energy values correspond to the thermodynamically preferred insertion of K atoms versus the formation of bulk K. The binding energies for all sites in VO₂ (M) are positive with regard to bulk K, which signifies that there are no suitable sites for K⁺ ions insertion. The only stable site

is the eightfold coordinated insertion site for VO₂ (B). However, it is still higher than most amorphous sites, indicating that VO₂ (a) is most conducive for K⁺ ions insertion due to the open structure of the amorphous configuration, which provides abundant K⁺ insertion sites for the effective transport of K⁺ ions.

## Synthesis and characterization of VO₂ cathode material

Different phases of VO₂ were synthesized, then varied characterizations were performed to check whether and how phase engineering improves the K storage performance of VO₂. The synthesis method can be found in the Supplementary information section in detail. The X-ray diffraction (XRD) pattern from VO₂ synthesized by citric acid without annealing exhibits only a broad peak around 25.5°, which is attributed to the CF. It indicates that this VO₂ is amorphous. For the materials synthesized by oxalic acid (with all raw materials expanded threefold), the peaks in XRD correspond to the crystalline VO₂ (B) phase (JCPDS No. 65-7960). Upon using oxalic acid and annealing, the sharp XRD peaks corresponding to the crystalline VO₂ (M) phase (JCPDS No. 82-0661) manifest upon the broad peak (Supplementary Fig. 1). Supplementary Fig. 2 illustrates the images for the several types of VO₂ at different magnifications. The field emission-scanning electron microscopy (FE-SEM) reveals the uniform growth of the different VO₂ phases, irrespective of the phase-type and distinct morphologies (Supplementary Fig. 2a, b, d, g). Supplementary Fig. 2c shows that the VO₂ (a) sample is composed of tiny particles with a diameter of 20 nm. The VO₂ (B) is rod-shaped with a thickness of 30 nm (Supplementary Fig. 2e) that grows perpendicular to the CF and forms a 1 μm thick film

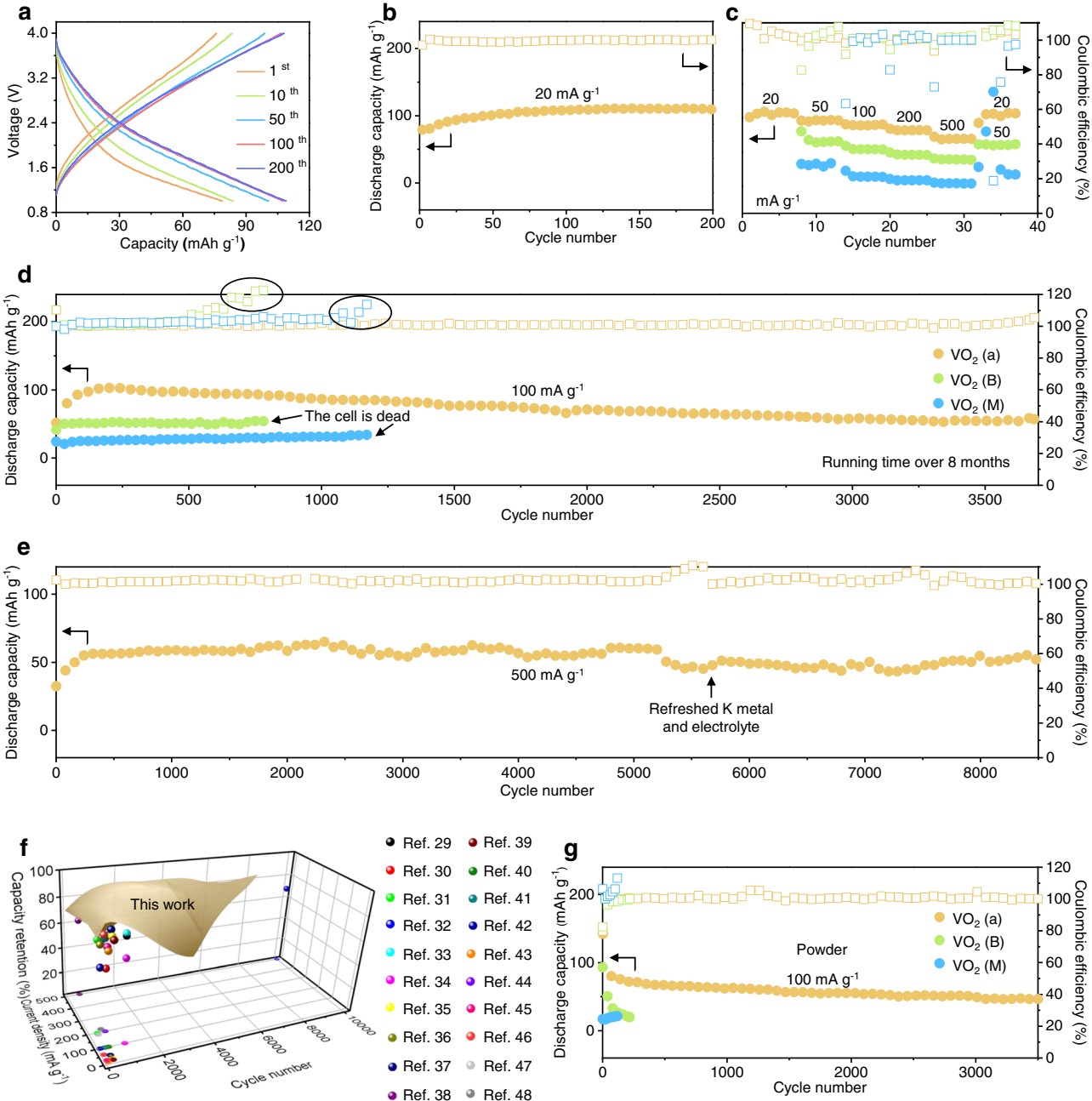

**Fig. 3 | Electrochemical performance optimization via phase engineering.**
**a** Discharge/charge curves and b) cycling performance of VO₂ (a) at a specific current of 20 mA g⁻¹. **c** Rate performance and (**d**) cycling performance of VO₂ (a), VO₂ (B), and VO₂ (M) at 100 mA g⁻¹. **e** Long cycle life performance of VO₂ (a) at 500 mA g⁻¹. **f** Comparison of cycling properties of VO₂ (a) with different types of PIB cathodes. **g** Cycling performance of the powders of amorphous VO₂, B phase VO₂, and M phase VO₂ at 100 mA g⁻¹.

(Supplementary Fig. 2f). On the other hand, the VO₂ (M) is nanoplate-structured with an average thickness of 30 nm and lateral size of about 300 nm, and 200 nm thick (Supplementary Fig. 2h, i). In particular, the energy dispersive spectroscopy (EDS) results of VO₂ (a) state in Supplementary Fig. 3 confirm the existence of V and O, and their ratios are very close to the chemical formula atomic ratio, as shown in Supplementary Table 1. The HRTEM image at atomic resolution and FFT pattern (Supplementary Fig. 4a, b) of VO₂ (a) are consistent with its amorphous nature and the structure constructed from AIMD simulations (Supplementary Fig. 4c). HRTEM characterizations were also performed on the VO₂ (B) and (M) samples, and the related images are shown in Supplementary Fig. 4d, h, respectively. Their corresponding FFT patterns along the [−130] and [−100] zone axes are shown in

Supplementary Fig. 4e, i, which are consistent with their simulated diffraction patterns (Supplementary Fig. 4f, j). The atomic structure models of VO₂ (B) and VO₂ (M) shown in Supplementary Fig. 4g and k are consistent with their HRTEM images.

## Effect of phase engineering on electrochemical properties
The discharge/charge curves of VO₂ (a) at low specific current (20 mA g⁻¹) are depicted in Fig. 3a. An increasing trend in capacity is observed initially, and the discharge capacity increases to 111 mAh g⁻¹ after 200 cycles (Fig. 3b). The rate performance of VO₂ (a) and crystalline VO₂ are compared over a wide specific current range. With increasing specific current from 20 mA g⁻¹ to 500 mA g⁻¹, the capacity of VO₂ (a) gradually decreases from 107 mAh g⁻¹ to 70 mAh g⁻¹ (Fig. 3c).

When the specific current is restored to 20 mA g$^{-1}$ after the high rate discharge/charge cycles, the average discharge capacity returns to 106 mAh g$^{-1}$. On the contrary, VO$_2$ (B) and (M) exhibit low discharge capacities of 39 and 5 mAh g$^{-1}$, respectively, at the extreme specific current of 500 mA g$^{-1}$. Next, the cycling performances of the VO$_2$ phases are illustrated in Fig. 3d for a specific current of 100 mA g$^{-1}$. The capacities of VO$_2$ (B) and VO$_2$ (M) are low (only 50 mAh g$^{-1}$ and 25 mAh g$^{-1}$, respectively), and their coulombic efficiencies gradually increase with cycle number, ultimately leading to cell death. Although the VO$_2$ (a) has an initial capacity of only 50 mAh g$^{-1}$, after 200 cycles, the discharge capacity of VO$_2$ (a) reaches 102 mAh g$^{-1}$ due to activation caused by penetration of the electrolyte. Notably, after 3700 cycles (over 8 months), the VO$_2$ (a) still retains a discharge capacity of 56 mAh g$^{-1}$. To further highlight the merits of VO$_2$ (a), long cycle performance tests at specific currents of 200 mA g$^{-1}$ (Supplementary Fig. 5) and 500 mA g$^{-1}$ (Fig. 3e) were performed. Interestingly, the VO$_2$ (a) preserves a highly stable coulombic efficiency with negligible degradation after 2500 cycles (compared with the maximum capacity at 200 mA g$^{-1}$). During cycling at 500 mA g$^{-1}$, the discharge capacity of VO$_2$ (a) reaches about 60 mAh g$^{-1}$ after 500 cycles. Its capacity drops to 52 mAh g$^{-1}$ after 8500 cycles, which translates into a capacity retention of 80%. The upward trend in the capacity of all cells in the initial cycles is attributed to activation caused by electrolyte penetration. Especially in amorphous structures, more active sites may be exposed for enhanced kinetics of the K-ion transfer reactions during cycling. The superior cycling stability of VO$_2$ (a) at different rates is represented by the shaded area in Fig. 3f, which is highly competitive compared to most previously reported PIB inorganic cathode materials (Prussian blue analogues[29–33], layered metal oxides[34–44], and polyanionic compounds[45–48]). To the best of our knowledge, the cycle life of PIB inorganic cathode materials is at most 1000 cycles, except for the defect-free potassium manganese hexacyanoferrate cathode material reported by Deng *et al*. in 2021[32]. To further verify the efficacy of amorphous VO$_2$ as PIB cathodes, the powders of amorphous VO$_2$, B phase VO$_2$, and M phase VO$_2$ were synthesized, which were confirmed by XRD (Supplementary Fig. 6). The cycle performance at 100 mA g$^{-1}$ of various phases of VO$_2$ powders are depicted in Fig. 3g, in which the amorphous VO$_2$ powder has more superior performance compared with other crystalline samples. The amorphous VO$_2$ powder, when directly coated on Al foil, exhibits faster electrolyte infiltration, and no activation process is required. After 3500 cycles, it maintains a discharge capacity of 46.3 mAh g$^{-1}$, which is slightly less than the VO$_2$ (a). As displayed in Supplementary Fig. 7a, b, the morphology of amorphous VO$_2$ powder is uneven and agglomerated. In addition, to eliminate the influence of CF substrate on electrode capacity, control experiments indicated that the capacity contribution from the pure CF substrate is negligible (Supplementary Fig. 8).

## K$^+$ (de)intercalation mechanism characterization and theoretical calculation

The K storage mechanism of VO$_2$ was investigated via ex situ X-ray photoelectron spectroscopy (XPS) in Supplementary Fig. 9. Initially, only V$^{4+}$ is detected in the V 2$p$ XPS region of various VO$_2$ phases, which exhibit two conspicuous peaks at 524.5 and 516.9 eV, corresponding to V$^{4+}$ 2$p_{1/2}$ and V$^{4+}$ 2$p_{3/2}$, respectively. After discharge to 1 V, the V$^{3+}$ peaks (2$p_{1/2}$: 522.5 eV and 2$p_{3/2}$: 515.0 eV) emerge in both VO$_2$ (a) and VO$_2$ (B), while the V$^{4+}$ peaks become weaker. Moreover, the peaks of V$^{3+}$ are not obvious in VO$_2$ (M), reassuring again that practically no K$^+$ guests are inserted into the VO$_2$ (M) host.

In situ XRD was performed to gain deeper insights into the K + (de) intercalation mechanism in the various VO$_2$ hosts. For VO$_2$ (a) illustrated in Supplementary Fig. 10a, it is evident that the XRD pattern remains unchanged during charging and discharging. No observable peaks appear concerning those of VO$_2$ in the entire XRD pattern,

demonstrating that the K$^+$ ions insertion into the VO$_2$ (a) does not alter the amorphous structure or generate a new crystalline structure. It should be noted that the peaks in the XRD pattern are the background peaks from the impurities of the metallic Be disc. A comparison test was conducted with a blank sample (Supplementary Fig. 11) to verify this conclusion. The ever-present amorphous structure results in excellent long-cycle performance and high capacity retention of the VO$_2$ (a). As shown in Supplementary Fig. 10b, the main peaks at 2$\theta$ = 25.3° and 33.6° can be indexed to the (110) and (310) peaks of the VO$_2$ (B) crystalline phase. During discharging, these peaks gradually shift to lower angles, indicative of lattice expansion due to the K$^+$ ions insertion into the VO$_2$ (B) structure and the electrostatic bond strength weakening[49]. Upon full charging to 4 V, the diffraction peaks gradually recover to their initial state, implying a reverse structural evolution. The in situ XRD of the VO$_2$ (M) was also characterized (Supplementary Fig. 10c). Its peaks at 27.8° and 37.1°, which represent the (011) and (200) lattice planes of the VO$_2$ (M), do not markedly shift during the discharging/charging, indicating that K$^+$ ions could not be inserted into the VO$_2$ (M) bulk. The K storage mechanism of VO$_2$ (a) and VO$_2$ (B) can be summarized as follows:

Discharge (potassiation):

$$VO_2 + xK^+ + xe^- \rightarrow KxVO_2 \tag{1}$$

Charge (de-potassiation):

$$KxVO_2 \rightarrow VO_2 + xK^+ + xe^- \tag{2}$$

DFT calculations were carried out further to understand the physical properties of VO$_2$ in different phases. Multiple non-equivalent diffusion paths for K$^+$ ions are present in the amorphous configuration. To ensure that the amorphous is more favorable than the crystalline phase, three long-range paths from different directions through the entire simulation cell were selected (Fig. 4a–c). The preferred diffusion directions in the crystalline structure were selected by the largest channel size present in the structure (Fig. 4d, e), and the relative migration energies are shown in Fig. 4f. The long-range disordered arrangement of atoms in the amorphous phase, which locally widens the channel size. The migration energies of all three paths of the amorphous phase are relatively small, which are 2.15, 2.74, and 1.85 eV, respectively. A diffusion barrier of 3.54 eV is found in VO$_2$ (B), for which the pathway for K$^+$ diffusion is mainly along the *b*-axis, as reported previously[50]. The typical diffusion pathway for K$^+$ transport in VO$_2$ (M) is within the open tunnel[28], and since the K$^+$ ion has no stable insertion site, its diffusion barrier is calculated for reference only. In addition to the diffusion barriers of K$^+$, the band structure was calculated. The main content of the total density of states (DOS) near the Fermi level is V 3$d$ and O 2$p$. Specifically, the pristine VO$_2$ (a) has a spin-polarized metallic electronic structure with no gap around the Fermi level in the DOS (Fig. 4g). The pristine VO$_2$ (B) is a spin-polarized semi-metal with a band gap of 0.29 eV (Fig. 4h), which is similar to previous reports[13,50]. It can be found that the K insertion does not noticeably affect the electronic structure of the VO$_2$ host both in VO$_2$ (a) and VO$_2$ (B). In contrast, the semiconductor phase VO$_2$ (M) shows a clear non-polarized DOS with a gap of 0.99 eV, which upon insertion of one K into the simulation cell, shows a smaller gap with spin polarization with a localized gap state (Fig. 4i). The above DOS results indicate that VO$_2$ (a) has better electronic conductivity, which brings about promising electrochemical performance. Combining the theoretical calculations with the experimental tests, it is concluded that phase engineering could effectively improve the ion mobility and electronic conductivity of PIB electrode materials.

## Influence of phase engineering on the kinetics

To further comprehend the storage process in the VO$_2$ electrode, electrokinetic analyses were conducted. Supplementary Fig. 12a, e, and i depict the cyclic voltammetry (CV) curves of VO$_2$ (a), VO$_2$ (B), and VO$_2$

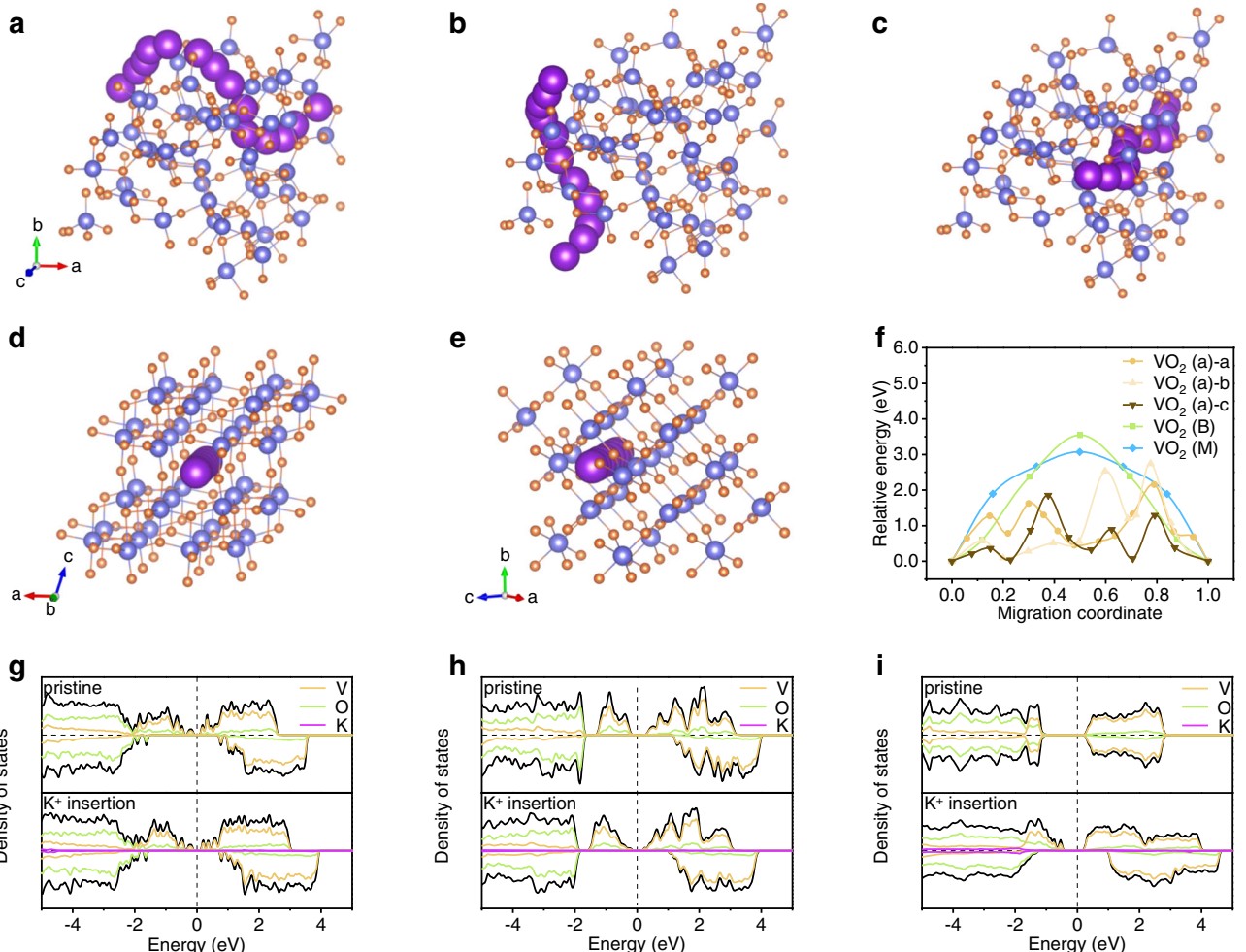

**Fig. 4 | DFT calculations of K storage.** The K migration paths of the **a–c** VO$_2$ (a), **d** VO$_2$ (B), **e** VO$_2$ (M). **f** Migration energy of K$^+$ ions for VO$_2$ in the three different phases. The DOS before and after K$^+$ insertion for **g** VO$_2$ (a), **h** VO$_2$ (B), **i** VO$_2$ (M).

(The partial density of states for the potassium is magnified by a factor of 200 for better visibility.).

(M) for scan rates from 0.1 mV s$^{-1}$ to 1.0 mV s$^{-1}$, respectively. The broadened redox peaks indicate that the storage mechanisms of K$^+$ ions consist of the diffusion and pseudocapacitive controlled redox processes. Easily adapting pseudocapacitive characteristics is a major advantage of vanadium-based cathodes[51]. The pseudocapacitive storage mechanism is expected to achieve both high energy and high power density through a fast redox reaction. In addition, when ions are inserted into the tunnels of the active material, intercalation pseudocapacitance occurs with no crystallographic phase change[52]. The degree of capacitive effect can be qualitatively analyzed by the value of the *b*-coefficient (Supplementary information). As shown in Supplementary Fig. 12b, the *b* value of VO$_2$ (a) is calculated to be 0.93, implying its predominantly pseudocapacitive kinetics. Moreover, the corresponding *b* values of VO$_2$ (B) and VO$_2$ (M) are 0.85 (Supplementary Fig. 12f) and 0.88 (Supplementary Fig. 12j), suggesting that although pseudocapacitance is still dominant, the contribution due to diffusion cannot be ignored. The high *b* value of VO$_2$ (M) may be due to the residual amorphous phase VO$_2$ after annealing at high temperatures. It can be observed that the CV curve area of VO$_2$ (M) per unit mass is minimal, representing a meager capacity. The capacitive contribution can be further quantified based on the current response (Supplementary information). With increasing scan rates from 0.1 mV s$^{-1}$ to 1.0 mV s$^{-1}$, the pseudocapacitance contribution for the total stored charge in VO$_2$ (a) increases from 83 to 94% (Supplementary Fig. 12c). These values are obviously higher than those in VO$_2$ (B) (from

55 to 79%, Supplementary Fig. 12g). As shown in Supplementary Fig. 12k, VO$_2$ (M) is affected by residual amorphous VO$_2$, and the proportion of pseudocapacitance is between that VO$_2$ (a) and VO$_2$ (B). Figures S12d, h, and l show the capacitive fractions at a scan rate of 1.0 mV s$^{-1}$ for the three phases. Such high pseudocapacitive contribution of VO$_2$ (a) is attributed to the open architecture with abundant structural defects, which provided more K$^+$ insertion sites and rapidly reversible reaction kinetics.

The migration kinetics of K$^+$ ions in the VO$_2$ cathodes were investigated by the galvanostatic intermittent titration technique (GITT). As shown in Supplementary Fig. 13a–c, during the first discharge/charge process in GITT, the VO$_2$ (a) exhibits the largest capacity and the smallest polarization, indicating the best kinetics for K$^+$ diffusion. The typical single-step of titrations for various phases of VO$_2$ are presented in Supplementary Fig. 13d–f, which were measured with a 600 s pulse duration and relaxed at the open circuit for 3600 s to reach the quasi-equilibrium state. The d$E$/d$\sqrt{t}$ shows the expected straight-line behavior (Supplementary Fig. 14a–c), from which the diffusion coefficient of K$^+$ ions ($D_k$) can be estimated (the exact method can be found in the Supplementary information section). The average $D_k$ for VO$_2$ (a) is about 10$^{-11}$ cm$^2$ s$^{-1}$ (Supplementary Fig. 14d), evidently higher than those for VO$_2$ (B) of 10$^{-12}$ cm$^2$ s$^{-1}$ (Supplementary Fig. 14e) and VO$_2$ (M) of about 10$^{-13}$ cm$^2$ s$^{-1}$ (Supplementary Fig. 14f). This comparison indicates the presence of more exposed K$^+$ channels in the amorphous configuration, facilitating rapid intercalation.

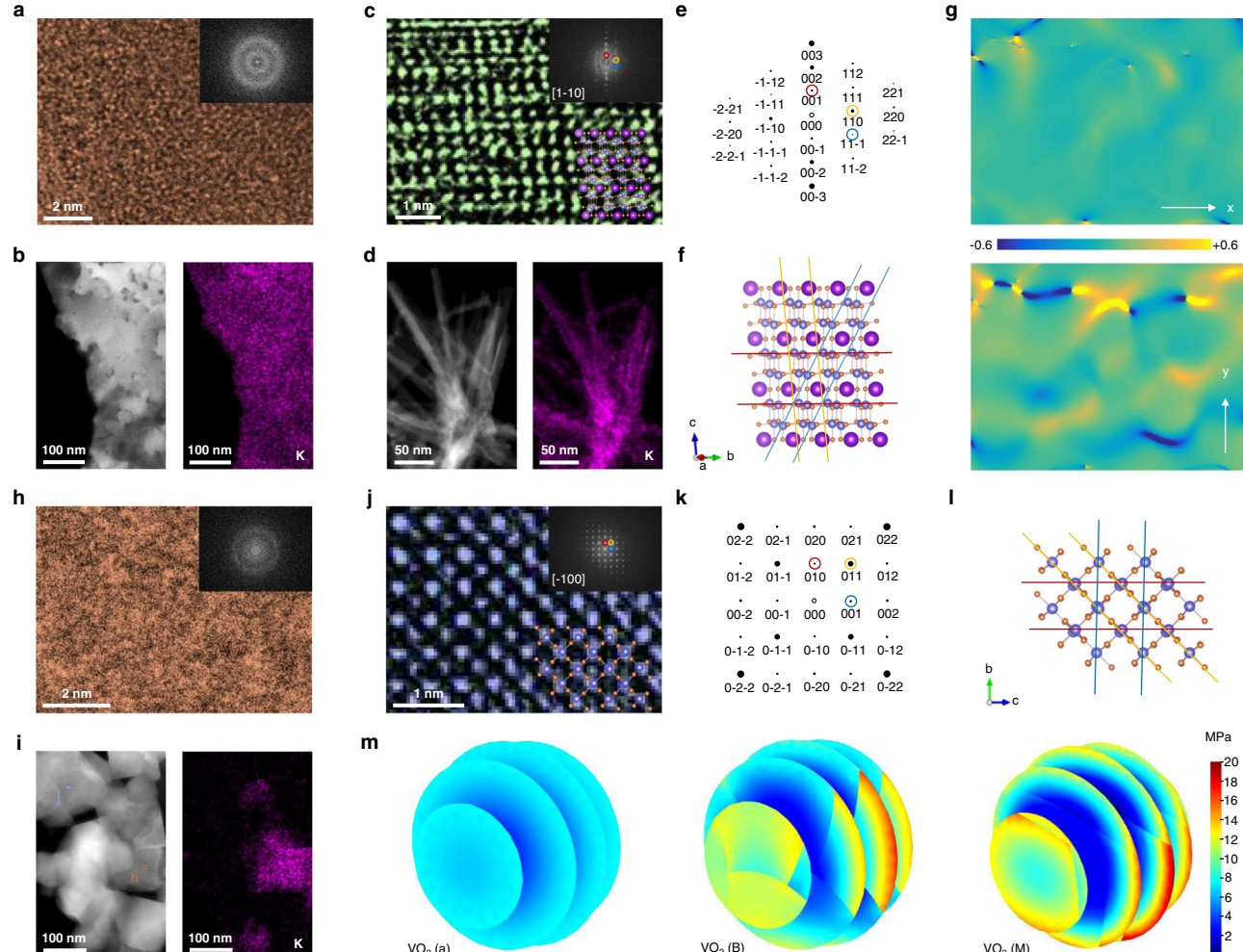

**Fig. 5 | Structural and stress analysis during cycling.** HRTEM images at atomic resolution and their related FFT patterns of **a** VO$_2$ (a), **c** VO$_2$ (B), and h) amorphous region of VO$_2$ (M) (the tan squared areas in Fig. 5i, j) crystalline region of VO$_2$ (M) (the dark blue squared areas in Fig. 5i) at discharged state after 100 cycles. HADDF-STEM images and corresponding elemental mappings of K for **b** VO$_2$ (a), **d** VO$_2$ (B), and **i** VO$_2$ (M). The simulated diffraction patterns of (**e**) VO$_2$ (B) and (**k**) VO$_2$ (M).

Schematic illustrations for the crystal structure of (**f**) VO$_2$ (B) and (**l**) VO$_2$ (M). **g** Strain mappings with uniaxial strain components $\varepsilon_{xx}$ (upper) and $\varepsilon_{yy}$ (lower), as obtained by GPA of the HRTEM image for VO$_2$ (B) after 100 cycles. **m** 3D view of equivalent stress after K$^+$ ions diffusion in VO$_2$ (a), VO$_2$ (B), and VO$_2$ (M) particles for 20 ms, respectively.

The electrochemical impedance spectroscopy (EIS) in the frequency range from $10^{-2}$ to $10^{5}$ Hz for VO$_2$ (a), VO$_2$ (B), and VO$_2$ (M) cells at open-circuit voltage states, charge state after 1 cycle, and 1000 cycles are shown in Supplementary Fig. 15. The charge transfer resistance R$_{ct}$ represented by the middle frequency semicircle in the three Nyquist plots is enormous at the initial state but decreases with increasing cycle number. The fitting results of impedance values are listed in Supplementary Table 2, and the R$_{ct}$ values of VO$_2$ (a) are always less than VO$_2$ (B) and VO$_2$ (M), indicating that the amorphous configuration has improved electronic conductivity.

**Investigation of the electrochemical-mechanical degradation mechanism**

Supplementary Fig. 16 shows the FE-SEM image of different VO$_2$ phases after 100 cycles. The morphology of VO$_2$ (a) still appears as uniform particles with no obvious change after cycling. At the same time, the crystalline phases show thick passivation films on their surfaces, which may lead to the attenuation of cycling performance. The HRTEM image at atomic resolution of VO$_2$ (a) at discharged state after 100 cycles is shown in Fig. 5a. As illustrated by its FFT pattern, the VO$_2$ (a) with inserted K$^+$ ions exhibits an amorphous structure after cycling. The scanning elemental mappings in Supplementary Fig. 17a and Fig. 5b

demonstrate the uniform distribution of V, O, and K in VO$_2$ (a). As evident in Fig. 5c, the VO$_2$ (B) still retains its crystalline structure with inserted K$^+$ ions after cycling. Simultaneously, all elements are also homogeneously distributed in Supplementary Fig. 17b and Fig. 5d. The simulated diffraction pattern (Fig. 5e) and the crystal structure (Fig. 5f) along [1 – 10] zone axis of VO$_2$ (B) are in good agreement with the FFT pattern in Fig. 5c. Geometric phase analysis (GPA) was also performed to determine the strain fields of VO$_2$ (B) (see Supplementary information section for details)[53]. The initial VO$_2$ (B) lattice parameter is selected as the reference, as exhibited in Supplementary Fig. 18, in which the compressive and tensile strains are negligible in either direction. As shown in Fig. 5g, after 100 cycles, a modest strain along the x-axis ($\varepsilon_{xx}$) appears, but an apparent strain along the y-axis ($\varepsilon_{yy}$) is present in the direction of inserted K$^+$ ions. This result implies that inserting K$^+$ ions in the crystal phase will cause certain lattice distortion and hinder ion migration. The situation for VO$_2$ (M) after 100 cycles at discharged state is consistent with the result after 1 cycle depicted in Fig. 1. The V and O elements are evenly distributed (Supplementary Fig. 17c). At the same time, K segregation appears in VO$_2$ (M) (Fig. 5i). The HRTEM image at atomic resolution and related FFT prove that the region with K distribution is amorphous (Fig. 5h), while the region without K is still crystalline (Fig. 5j). From the simulated diffraction

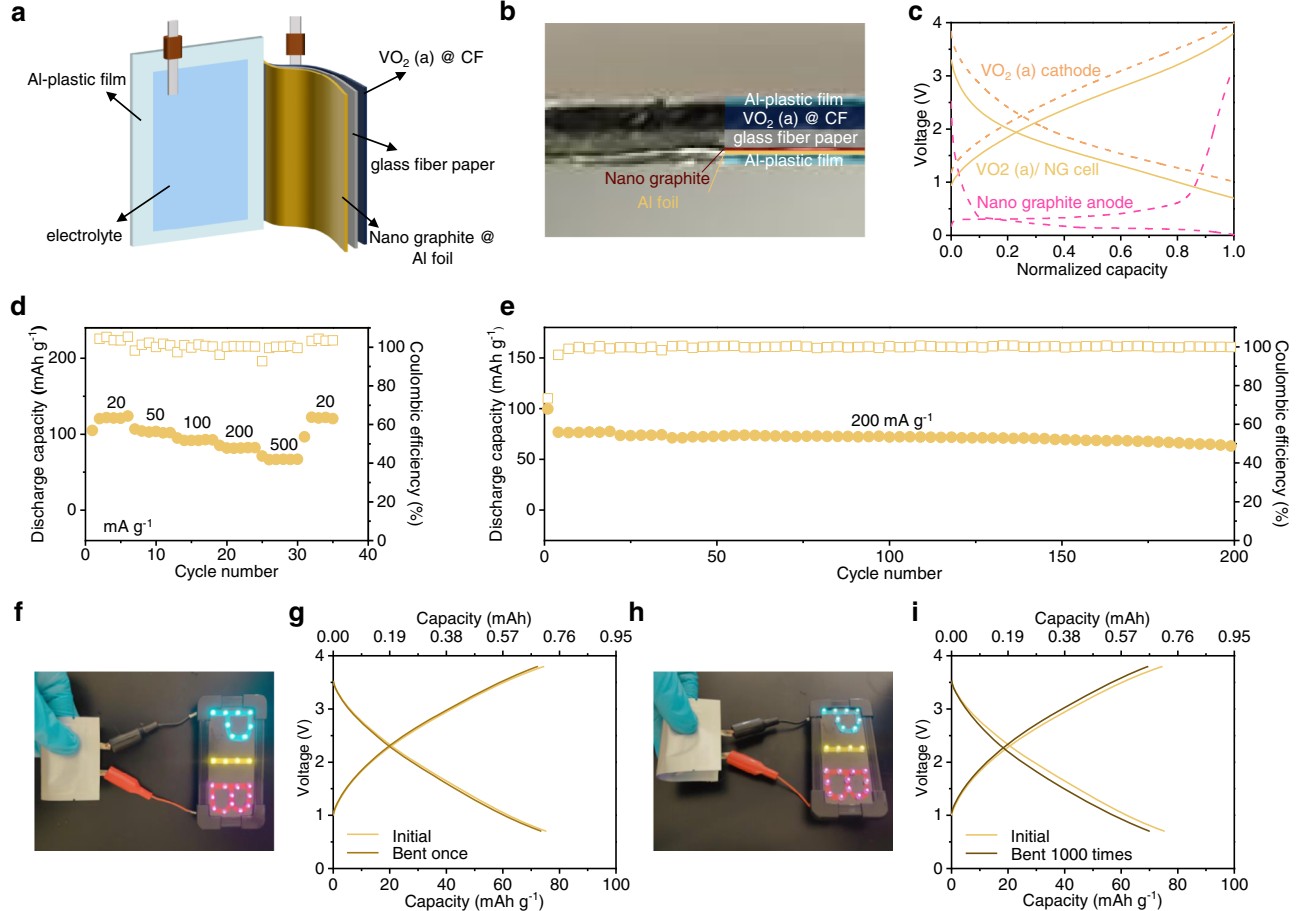

**Fig. 6 | The electrochemical performance of the full battery. a** Schematic diagram of the K ion full battery. **b** An optical photograph of the cross-section of the soft-pack battery. **c** Discharge/charge profiles of the half cells and the full battery. **d** Rate and (**e**) cycling performance at 200 mA g⁻¹ for VO₂ (a)/nano-graphite in full battery. Image of LEDs powered by a soft-pack battery composed of VO₂ (a) after bending (**f**) once and (**h**) 1000 times. Discharge/charge profiles of the second cycle after bending (**g**) once and (**i**) 1000 times compared to the initial.

patterns and the crystal structure along [−100] zone axis (Fig. 5k, l), it is clear that the crystalline structure is the M phase of VO₂.

The electrochemical-mechanical degradation mechanism of amorphous and crystalline phases, as well as the stress change in them, were studied by diffusion-induced stress model[54]. To explain more reasonably, the stress developed in the three phases of VO₂ for the same K⁺ ions diffusion time of 20 ms was compared (Fig. 5m). Uniform stress distribution is caused by the atomic disorder and isotropy in the VO₂ (a) particles, which leads to small variations in the stress distribution over the particle (Supplementary Fig. 19a) and ensures excellent cycling performance. By comparison, the anisotropy caused by the long-range ordered arrangement of atoms in the VO₂ (B) generates uneven stress changes in the particles with K⁺ ions inserted, which readily leads to high-stress concentrations near the grain boundary interface and hinders ion diffusion (Supplementary Fig. 19b). Because there is no stable K-insertion site in the VO₂ (M), stress is caused only if K⁺ ions are forced into the M phase, which can also be inferred from the large von Mises strain stress curve along the radial direction of spherical particles as shown in Supplementary Fig. 19c. The stresses were also computed when K⁺ ions sufficiently inserted into the VO₂ (B) and VO₂ (M) phases (Supplementary Fig. 20), and their stresses are extremely large. Supplementary Fig. 21 illustrates the schematic structural changes for VO₂ (a) and VO₂ (B) during the potassiation and depotassiation. As cycles proceed, the VO₂ (B) strain intensifies, leading to induced polarization and a sharp decline in cycling performance. Even if K⁺ ions are removed from the lattice, the structure does not return to its original state. By comparison, VO₂ (a)

can retain its original structure after cycling. The open architecture tolerates volume variation more uniformly, and the stable structure facilitates reversible K⁺ ions insertion/extraction.

## Electrochemical performance of the full battery

Benefiting from the flexible feature of the CF, the assembled soft-packaged battery using VO₂ (a) is likewise flexible. The schematic diagram of the K ion full battery is displayed in Fig. 6a. For more intuitive, an optical photograph of the cross-section of the soft-packaged battery is shown in Fig. 6b, which is assembled by sandwiching the fiberglass paper and electrolyte between the VO₂ (a) cathode and pre-potassiated nano-graphite anode. The typical discharge/charge curves of VO₂ (a) and nano-graphite electrodes in half cells and the full battery are shown in Fig. 6c. Figure 6d presents the rate performance of the VO₂ (a)/nano-graphite in full battery. It delivers a reversible capacity of 119 mAh g⁻¹ to 67 mAh g⁻¹ when the specific current increases from 20 mA g⁻¹ to 500 mA g⁻¹ (based on the mass of the cathode active materials). The cycling study carried out at a current of 200 mA g⁻¹ indicates that 63 mAh g⁻¹ is retained after 200 cycles with approximately 100% coulombic efficiency (Fig. 6e). The full battery performance with VO₂ (a) as the cathode is compared with that of the reported systems (Supplementary Fig. 22a, b). The VO₂ (a) displays one of the best battery performances, exhibiting good rate capability and cycling performance. Figure 6f shows that the LEDs continue to light when the soft-packaged battery is bent at a large angle. The discharge/charge profiles of the second cycle are shown in Fig. 6g, which is the same as before bending. After the battery is bent 1000 times, the LEDs light still glows (Fig. 6h),

and the discharge/charge curves are not significantly different from the initial ones (Fig. 6i). Furthermore, the FE-SEM images of the flexible electrode after multiple bends (Supplementary Fig. 23) also show essentially the same morphology as the initial state, revealing its excellent potential as a flexible energy storage device.

## Discussion

In summary, this study shows that phase engineering could effectively improve the K storage performance of $VO_2$. The elemental map shows that $K^+$ ions could be evenly inserted into the amorphous structure. The DFT calculations likewise confirm that $VO_2$ (a) could provide more stable $K^+$ insertion sites compared to crystalline $VO_2$ (B) and $VO_2$ (M), leading to a high capacity of 111 mAh g$^{-1}$ at 20 mA g$^{-1}$. In addition, combined in situ XRD characterization with the equivalent stress simulation, it can be seen that $VO_2$ (a) always maintains an amorphous structure during cycling, the stress is negligible, and it can exhibit an excellent long cycle life. It operated for 8 months at 100 mA g$^{-1}$ and 8500 cycles at 500 mA g$^{-1}$ with a capacity retention of 80%. On the other side, the crystalline $VO_2$ (B) suffers from comparatively large stress obtained by strain mapping analysis of HRTEM images, and consequently, its structure got damaged during cycling. The crystalline $VO_2$ (M) does not even stably insert $K^+$ ions. In addition, through DFT calculations, the $VO_2$ (a) exhibits better electronic conductivity and a lower $K^+$ ions migration energy barrier, which also contributes to its better electrochemical performance. Furthermore, the diffusion kinetics of $VO_2$ (a) was systematically studied, and pseudocapacitance behavior plays a more prominent role in the capacity contribution. Its calculated $D_k$ is 1 to 2 orders of magnitude higher than the crystalline phases. This study provides promising perspectives for electrode materials used in rechargeable alkali batteries.

## Methods

### Materials synthesis

The $VO_2$ (a) was prepared by hydrothermal method. 1 mmol $V_2O_5$ (99.6%; Sigma-Aldrich) and 3.3 mmol citric acid (99.0%; Sigma-Aldrich) were added to 10 ml distilled water, followed by stirring at 80 °C for 30 min. About 2.5 mL $H_2O_2$ (30 wt%; Aladdin) was added dropwise to the previous solution and stirred, then added 25 ml alcohol and a piece of CF (CeTech) and transferred to a 50 ml Teflon-lined stainless steel autoclave, which was maintained at 180 °C for 3 h. Lastly, the product was washed thoroughly with deionized water and ethanol and dried under vacuum at 60 °C for 12 h. $VO_2$ (M) was obtained using the same process as $VO_2$ (a), except that the citric acid was replaced by oxalic acid (99.0%; Sigma-Aldrich) and annealed at 500 °C for 4 h under Ar atmosphere. $VO_2$ (B) was also made from oxalic acid, but compared with $VO_2$ (a), the amount of all raw materials was increased threefold[20,55]. Since the synthesis method mentioned above could not obtain the individual amorphous $VO_2$ powder sample, 3 mmol $VOSO_4 \cdot xH_2O$ (99.9%; Aladdin) and 6 mmol NaOH (97.0%; Aladdin) were stirred into deionized water for 1 h. The precipitates were centrifuged with water and ethanol and dried at 80 °C for 12 h. Next, the product was calcined at 300 °C for 8 h to obtain the amorphous $VO_2$ powder[8]. The B phase and M phase $VO_2$ powder were prepared in the same way as $VO_2$ (B) and $VO_2$ (M), but without adding the CF.

### Materials characterization

The XRD patterns were recorded on a Bruker D8 ADVANCE X-ray diffractometer with Cu Kα irradiation (λ = 1.54186 Å) between the angular range of 5° to 80° (2θ). The metallic Be disc, test electrode, separator, potassium counter electrode, and electrolyte were assembled into a customized test model for in situ XRD measurements. The in situ XRD measurements used a small specific current (20 mA g$^{-1}$ or 50 mA g$^{-1}$) with a voltage window of 1 V – 4 V. The morphology of the samples was characterized by FE-SEM (Jeol JSM-7610FPlus) at 5 kV with EDS analysis, and the microstructure of the samples was identified by STEM (Thermo

Scientific Talos F200X) at 200 kV with elemental mappings analysis. The valences of the different $VO_2$ samples were analyzed by XPS (SHIMADZU AXIS SUPRA +). The density of the amorphous $VO_2$ powders was measured by the automatic true density tester (AccuPyc II 1340).

### Electrochemical measurements

Electrochemical properties of $VO_2$ in different phases were evaluated by assembling 2025 coin-type cells in a highly pure argon-filled glove box ($O_2$ and $H_2O$ levels less than 0.1 ppm) with the as-prepared freestanding samples as the working electrode, a glass microfiber paper (diameter: 19 mm, thickness: 260 μm, Whatman) as the separator, and a solution of 2.5 M KFSI (98.0%; SongJing) in triethyl phosphate (99.5%; SongJing) as the electrolyte. In addition, the loading mass of the active material was approximately 0.8 mg cm$^{-2}$, and the diameter of the electrode was 12 mm. The coin was assembled with K metal (99.0%; Aladdin) as the counter electrode for the half cell. The K metal electrode is used by rolling the potassium block into a thin slice with about 100 μm and then stamping it into a circular electrode with a diameter of 12 mm, without any other special treatment. For the full battery, the pre-potassiated nano-graphite (XFNANO) was used as an anode, whose slurry was prepared by 80 wt% active material, 10 wt% conductive carbon black (Super P), and 10 wt% CMC onto an Al foil. The nano-graphite anode was first assembled with K metal electrode into a half cell for 5 cycles and discharged to 0.001 V to eliminate its irreversible capacity loss before matching the full battery. The size of the full battery cathode is 3 × 4 cm, and the anode and the separator are slightly larger than that. The N/P ratio of the anode to cathode materials was about 1.05, and the capacity of the full battery was calculated based on the mass of $VO_2$. The full battery used the same electrolyte as the half cell, and its injection volume was about 0.4 ml. When measuring the cycle performance of pure CF, the capacity was calculated based on the actual mass itself, and the current was obtained empirically from the other electrodes with active materials, which was about 0.04 mA. The galvanostatic discharge/charge, cycling, rate, and GITT tests were carried out using the Neware BTS-53 system. The voltage range was 1 to 4 V (vs. K/K$^+$) in the half cell, while 0.7 to 3.8 V was for the full battery. All electrochemical energy storage tests were carried out in an incubator at 28 ± 0.5 °C. The CV curves and the Nyquist plots were recorded using a CHI660e electrochemical workstation at the scan rate of 0.1 to 1 mV s$^{-1}$ and the frequencies ranging from 10$^{-2}$ to 10$^5$ Hz, respectively.

### Detailed analysis of CV

Based on the peak currents at each scan rate, the surface-controlled and diffusion-controlled processes can be quantitatively concluded:

$$i = a v^b \tag{3}$$

The peak current $i$ is usually proportional to the sweep rate $v$ when the process is purely capacitive, and the adjustable parameters value of $b = 1$. Similarly, the process is that of pure diffusion when $b = 0.5$. The $b$ value can be calculated from the slope of the log ($i$) versus log ($v$) plot. The capacity can be divided into the competitive surface-controlled ($k_1 v$) and diffusion-controlled ($k_2 v^{1/2}$) parts depicted by the following equation[56]:

$$i = k_1 \nu + k_2 \nu^{1/2} \tag{4}$$

### Detailed description of GITT

The chemical diffusion coefficient $D_k$ was calculated based on the following equation[57]:

$$D_k = \frac{4}{\pi} \left( \frac{m_B V_M}{M_B S} \right)^2 \left( \frac{\Delta E_s / \tau}{dE/d\sqrt{t}} \right)^2 \tag{5}$$

where $m_B$, $M_B$, and $V_M$ are the mass, atomic weight, and molar volume of the sample, respectively. $S$ is the cross-sectional area common to the electrolyte and the sample electrode, $\tau$ is the pulsed current time, $\Delta E_s$ is the change of the quasi-equilibrium voltage, $dE/d\sqrt{t}$ is directly obtained from the measured voltage as a function time during the constant current pulse[58]. As shown in Supplementary Fig. 14a, b, and c, the $dE/d\sqrt{t}$ is the straight line. Equation (5) can be simplified as follows:

$$D_k = \frac{4}{\pi\tau}\left(\frac{m_B V_M}{M_B S}\right)^2\left(\frac{\Delta E_s}{\Delta E_t}\right)^2 \tag{6}$$

where $\Delta E_t$ is the total change of the cell voltage $E$ during the current pulse, neglecting the IR drop.

## Detailed of the simulated diffraction pattern

The simulated diffraction pattern was performed with the tool cell-Viewer in CrysTBox software. This tool can show the material from different perspectives in the direct atomic lattice (cell view) and reciprocal lattice (diffraction view). The only required input of this tool is the CIF of the sample material. In the theoretical spot diffraction pattern, individual diffraction spots are represented by disks. The darker the spot appears, the better satisfied the diffraction condition is. Crosses represent extremely weak or forbidden reflections[25].

## Detailed of GPA

The strain analysis was performed with the tool gpaGUI by CrysTBox[59]. The method transforms the input HRTEM image into an artificial diffractogram using a fast Fourier transform (FFT). Computing the strain requires two noncollinear reflections to be selected.

## Computational methods

All the density functional theory (DFT) calculations were implemented using the Vienna ab initio Simulation Package (VASP). The projector augmented wave (PAW) pseudopotential was employed to describe the ionic cores, and Perdew-Burke-Ernzerhof (PBE) formulation was used to describe the generalized gradient approximation (GGA) exchange-correlation function with a kinetic energy cutoff of 520 eV[60–64]. A rotationally invariant Hubbard correction was applied, as proposed by Dudarev et al. with (U-J) = 3.1 eV for V d states[65]. The ab initio Molecular Dynamics (AIMD) simulation was carried out to study the amorphous structures of $VO_2$ with a $1 \times 1 \times 1$ $k$-point grid of Monkhorst-Pack. The convergence criteria for the energy and force are $10^{-4}$ eV/cell and 0.05 eV/Å, respectively. The AIMD simulations were performed at 300 K in the Nosé-Hoover isokinetic ensemble with an equilibration time of 10 ps and a time step of 1 fs. The initial density of the amorphous simulation structure was about 2.8 g cm$^{-3}$, which agreed with the measured amorphous phase $VO_2$ density of 2.81 g cm$^{-3}$ (Supplementary Table 3). The structure provided an average coordination number of 5.08 for V (with a cutoff for the V-O bond of 2.4 Å), close to the measured value of 4.8[66]. The amorphous structure is less stable than the crystalline phase by 0.22 eV per atom. The equilibrium lattice constants were optimized utilizing a $2 \times 2 \times 4$, $2 \times 3 \times 2$, and $2 \times 3 \times 5$ Monkhorst-Pack $k$-point grid for Brillouin zone sampling of $VO_2$ (a), $VO_2$ (B), and $VO_2$ (M) unit cell, respectively. The Gaussian smearing method with a width of 0.05 eV was applied with convergence criteria of $10^{-5}$ eV/cell and 0.02 eV/Å for energy and force, respectively. The K$^+$ ions migration barriers were calculated using the Climbing Image-Nudged Elastic Band method[26].

## Simulation methods

The finite element method by COMSOL Multiphysics software package was used to calculate the material stress through potassium ions diffusion induction[54]. A spherical model with a diameter of 15 nm was constructed according to the actual grain size, in which the amorphous phase was considered to be isotropic, while the crystalline phase was considered to be anisotropic. The developer of the built-in APP generated random grain boundaries, and the crystalline sphere was divided into several parts with different orientations. The diffusion stress induced by the insertion and extraction of potassium ions was calculated by thermal analogy using solid mechanics and dilute matter transfer modules. The tensor $D$ denotes the diffusion coefficient. We set $D_a = 1 \times 10^{-11}$ cm$^2$ s$^{-1}$, $D_B = 1 \times 10^{-12}$ cm$^2$ s$^{-1}$, and $D_M = 1 \times 10^{-13}$ cm$^2$ s$^{-1}$. The constitutive equations for stress and strain are the following:

$$\varepsilon_{ij} = \varepsilon_{ij}^e + \varepsilon_{ij}^c = \frac{1}{E}\left[(1+\nu)\sigma_{ij} - \nu\sigma_{kk}\delta_{ij}\right] + \beta_{ij}c \tag{7}$$

$$\sigma_r = \lambda e + 2\mu\varepsilon_r - \frac{1}{3}(3\lambda + 2\mu)\Omega(C - C_0) \tag{8}$$

$$\sigma_\theta = \lambda e + 2\mu\varepsilon_\theta - \frac{1}{3}(3\lambda + 2\mu)\Omega(C - C_0) \tag{9}$$

$\varepsilon_{ij}$ and $\sigma_{ij}$ are the stretch and stress tensors, respectively. $E$ is Young's modulus, and $\nu$ is Poisson's ratio. Here the approximate values of Young's modulus (270 GPa)[67] and the Poisson ratio (0.243)[68] were utilized. The diffusion of potassium ions can be depicted by the modified Fick's law, which involves the effect of stress on diffusion:

$$\frac{\partial c}{\partial t} + \nabla \cdot \left(-D_{ij}\nabla c + \frac{D_{ij}c}{RTc_{ref}}\nabla\sum\beta_{ij}\sigma_{ij}\right) = 0 \tag{10}$$

$$\mu_s = \mu_0 + RT\ln C - \Omega\sigma_m \tag{11}$$

where $D_{ij}$ is the diffusion coefficient tensor, $R$ is the universal gas constant, and $T$ is the temperature.

## Data availability

The data that support the findings of this study are presented in the manuscript and supplementary information, or from the corresponding authors upon reasonable request. Source data are provided with this paper.

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

## Acknowledgements

This work was financially supported by the National Natural Science Foundation of China No. U20A20247 and 51922038. AMR acknowledges the financial support through the RA Bowen Endowed Professorship. The authors also acknowledge Zhixiang Si at Guangdong Technion–Israel Institute of Technology for contributing to the TEM experiments.

## Author contributions

BL supervised the entire project. LW planned, performed the experiments, and wrote the manuscript. HF, and KG contributed to the DFT calculations. SL assisted in data analysis. JZhu, JZhou, SW, and AMR were all involved in the discussions. LC conducted STEM characterizations. All authors discussed the results and commented on the manuscript.

## Competing interests

The authors declare no competing interests.
