## [Peer Review File · Nature Communications]

REVIEWER COMMENTS

Reviewer #1 (Remarks to the Author):

This work reported K-ion storage in the amorphous VO₂ (designated as VO₂-a) materials, showing a reasonable capacity performance of 111 mAh/g at 0.02 A/g. Although this work strongly integrates theoretical and experimental approaches to understand the structure of amorphous VO₂, in addition to in situ XRD and equivalent stress measurement, the research originality, to my knowledge, may not meet the caliber of Nature Communications for the following reasons:

1. VO₂-a is a highly disordered material, only showing one broad diffraction peak (Figure S2). The determination of the crystalline phase is very challenging, and so is the construction of the simulation model and the interpretation of in situ XRD.
2. very strong pseudocapacitive charge storage behavior of VO₂-a (Figure S11), making it less likely a candidate for high-capacity cathode materials for K-ion battery
3. the data quality could be better (e.g., XPS, XRD).

Reviewer #2 (Remarks to the Author):

The manuscript presents a combined experimental-computational study of the use of VO₂ as a potassium ion battery cathode. The key finding is that amorphization facilitates K storage. Vanadia phases continue to be promising for beyond-Li batteries and as vanadia present rich phase diagrams, phase selection is an important issue. Therefore this is in principle impactful and publishable if the work is done right. The following issues need to be addressed to ensure that:

- The authors present a combined experimental-DFT study but refer only to experimental prior work. There is a relatively long line of computational research arguing for the benefits of amorphization for the storage of large(er than Li) cations where crystalline phases do not work well. This was rationalized earlier in DFT studies on Si and titania in J. Phys. Chem. C 119, 13496–13501 (2015), J. Power Sources, 278, 197-202 (2015), Comput. Mater. Sci., 94, 214-217 (2014) which were later borne out experimentally. Specifically promise of amorphization for K storage was shown in MRS Advances 1, 3069–3074 (2016) on the example of carbon. Regarding relative performance of different vanadia phases, a comprehensive review of vanadia phases for use in metal ion batteries was published in J. Phys. D 3, 083001 (2020) that conceptualizes many of the issues the authors are dealing with. The present paper would be well served if it contextualized itself better in the light of those prior research trends that it continues and which solidified the case for using amorphous phases.
- The authors appear to report the lowest diffusion barrier in the amorphous phase, however, this does not guarantee that long-range diffusion is more favored vs a crystal, as in the a-VO₂ there will be multiple non-equivalent diffusion barriers. One needs to show a long-range path (through the entire simulation cell)
- Why was Grimme D3 used?? These are not layered phases, so dispersion corrections are not needed and will in fact make the accuracy worse. Moreover, the Grimme correction should not be applied to K as it is fully ionized (= no valence shell to participate in dispersion interactions). This issue could throw off the insertion energies by tenths of an eV.
- the parameters of the generated amorphous structure should be given and compared to experimental a-VO₂ (density, energy metastability, coordination number statistics)
- the sentence "The conjugated gradient method was applied with a smearing width of 0.15 eV" does

not make much sense. Do the authors mean smearing of electronic occupancies?

- Figs 1 and 3 are too crowded. It is better to split the exp and the models into different figures.

- the authors refer to the same K positions K1, K2, K3 in the crystalline and amorphous case. But in a-VO₂ there will be multiple and different insertion sites, certainly more than 3 (if the amorphous structure is realistic). Insertion energies in a representative set of such sites should be given.

- something appears to be not right with the PDOS. At that K concentration are those? PDOS should be shown before and after K insertion. The semiconductor phases such as M should show a clear non-polarized DOS with a gap, which upon insertion of one K into the simulation cell should show metallic character with spin polarization with either a conduction band delocalized state (without +U) or a localized gap state (with +U which should be applied in this case but apparently is not). What is shown is something else.

The above suggests the need for for a major revision.

Reviewer #3 (Remarks to the Author):

In this manuscript, the application of phase engineering has been realized by constructing an amorphous phase VO₂ cathode, and the surprising effects have been achieved in potassium ion batteries. The idea of this paper is clear, especially the use of a large number of TEM characterizations provides a good confirmation of the effect of the phase engineering proposed by the authors. Compared with the crystalline phase, the amorphous phase VO₂ not only has stable potassium storage sites, but also shows little deformation during battery use. The analysis of the HRTEM is exact, such as the analysis of the lattice, the diffraction pattern, and the zone axis. This study establishes a relationship between phase and electrochemical properties, which is of great importance. Hence, I would be glad to recommend the acceptance of this paper in Nature Communications after addressing the following minor issues.

(1) Please explain in detail how the capacity of pure carbon cloth is measured in Supplementary Figure 8.

(2) Additional experimental details on the full battery assembly should be provided, such as the proportion of each component on the negative electrode and the amount of electrolyte injection.

(3) Amorphous VO₂ has no characteristic peak in XRD characterization. In this paper, the amorphous sample is identified as VO₂ based only on the XPS results. Additional evidence is needed, such as the results on the ratio of elements in the EDS.

(4) The authors should carefully check the main text to ensure no errors. For example, "The DFT calculations likewise confirmed that..." should be "The DFT calculations likewise confirmed that...".

(In Discussion)

(5) Authors need to read more articles about phase engineering carefully and cite them selectively. Do the results of this paper have a generality to lithium ion batteries?

Reviewer #4 (Remarks to the Author):

Phase engineering is an effective strategy to regulate the properties, functions, and applications of nanomaterials by rationally tuning their atomic arrangements. In this work, at the atomic level, the authors found that the crystalline region in the M phase VO₂ is not able to store potassium, while the residual amorphous region remaining from the annealing preparation has a stable potassium intercalation ability, which is an interesting phenomenon. The authors first realized the application of VO₂ as a PIBs cathode by using an amorphous phase, demonstrating the effectiveness of phase engineering. Moreover, the amorphous VO₂ shows an excellent sustained cycling performance of 8500 cycles. Overall, this is a meaningful study that sheds light on the exploration of high-

performance rechargeable batteries and is suitable for publication in Nature Communications. The authors need to make the following revisions before publication.

1. The direction of the coordinate axes is not given in Fig. 1h, is it shared with Fig. 1i?
2. Please include the method and description of the simulated diffraction pattern in the supplementary information.
3. Fig. 5g and i are suggested to use the double abscissa, the lower x-axis is the capacity per unit mass, and the upper x-axis is the real capacity of the full battery.
4. In Fig. 5f and h, the author can try to drive something with greater power, such as a light plate composed of multiple LEDs, which is more convincing than lighting such a single small bulb.
5. It is suggested to supplement the SEM morphology characterization of the flexible electrode after multiple bending, and compare it with the initial state.
6. The full name of the XPS analysis is not provided anywhere, only the abbreviation. Please add the full name of the characterization upon the first usage of the abbreviation.

Point-by-point response to the reviewers' comments

We thank the reviewers for their valuable comments, which we found very constructive and valuable. We have addressed all the comments from the reviewers to the largest possible extent. For ease of reference, our responses are shown in blue, and the resulting changes to the manuscript are in red.

Reviewer #1 (Remarks to the Author):

This work reported K-ion storage in the amorphous VO₂ (designated as VO₂-a) materials, showing a reasonable capacity performance of 111 mAh/g at 0.02 A/g. Although this work strongly integrates theoretical and experimental approaches to understand the structure of amorphous VO₂, in addition to in situ XRD and equivalent stress measurement, the research originality, to my knowledge, may not meet the caliber of Nature Communications for the following reasons:

Response: We thank the reviewer for acknowledging that our work “strongly integrates theoretical and experimental approaches.” Moreover, we appreciate the opportunity to clarify the originality of our work.

(1) First, there are only a few combined theoretical and experimental studies for amorphous cathodes. For a long time, most cathode materials investigated for rechargeable batteries were crystalline. For example, layered oxide materials and olivine-type lithium iron phosphate materials are used in lithium-ion batteries, while

Prussian blue materials, layered oxides, and polyanionic materials are in sodium and potassium ion batteries. However, these cathode materials are limited by their inherent structural dimensions, and ion transport is difficult. Especially for ions with large ionic radii like potassium ions, they undergo large deformation during charging and discharging, resulting in poor electrochemical performance. In contrast, the phase engineering technology reported in this manuscript, viz., amorphous phase VO₂, showed better performance than the corresponding crystalline phase with significantly lower mechanical strain.

(2) Secondly, the low electronic conductivity and unstable structure limit the rate and cycling performance of vanadium-based materials. This study, for the first time, utilized phase engineered VO₂ as potassium ion battery cathodes and achieved excellent electrochemical performance. To the best of our knowledge, this is the longest cycling among the metal oxide cathodes of potassium ion batteries, and the capacity retention is excellent, viz., 80% after 8500 cycles.

(3) Thirdly, in our study, we characterized the potassium storage in the amorphous and crystalline phases of VO₂ at the atomic scale using STEM, which is a significant advance as it established the relationship between phases and electrochemical properties, and provided a good confirmation for the validity of phase engineering.

(4) Finally, the stability of cathode materials for rechargeable batteries with large ionic radii is always problematic, as their structure can be compromised during cycling due to constant stress, resulting in capacity loss. This study strongly integrated theoretical and experimental approaches to elicit unprecedented electrochemical performance of

VO₂ (a), providing a new idea for selecting cathode materials for future rechargeable batteries, such as K, Zn, and Al ion batteries.

1. VO₂-a is a highly disordered material, only showing one broad diffraction peak (Figure S2). The determination of the crystalline phase is very challenging, and so is the construction of the simulation model and the interpretation of *in situ* XRD.

Response: We appreciate this concern raised by the reviewer. The amorphous phase VO₂ (a) is indeed a highly disordered material. By means of HRTEM (Supplementary Fig. 4), XRD (Supplementary Fig. 1), and *in situ* XRD (Supplementary Fig. 10), we proved beyond a doubt that we achieved our research object of synthesizing the amorphous VO₂ (a).

In addition, the construction of its molecular model, as mentioned by the reviewer, is indeed challenging. We used the *ab initio* Molecular Dynamics (AIMD) simulation method to build a model of the amorphous VO₂ (a). The simulated amorphous structure agrees with the experimentally generated structures in metrics such as mass density and coordination number, which fully proved the correctness of our constructed model. On this basis, we performed calculations of binding energy, migration energy, and density of states, and successfully confirmed the superiority of the amorphous VO₂ (a).

We also characterized the amorphous VO₂ (a) by *in situ* XRD, and found that the insertion of K⁺ ions into VO₂ (a) does not change the amorphous structure or create a new crystal structure (Supplementary Fig. 10a). In other words, the VO₂ (a) always remained amorphous and yet ensured excellent long-cycle performance and high capacity retention. In contrast, crystalline B-phase VO₂ when characterized by *in situ*

XRD exhibited a shift in the position of its characteristic peaks during charging and discharging (Supplementary Fig. 10b). Although M-phase VO₂ cannot store potassium, its characteristic peaks could still be seen by *in situ* XRD. The XRD data for the B- and M-phase VO₂ are significantly different from those of the amorphous phase.

Our ability to accurately characterize the amorphous VO₂ phase and elicit unprecedented electrochemical properties precisely illustrates the importance of this study.

Revision: (Computational methods part of Supplementary information) The initial density is about 2.8 g cm⁻³, which agrees with the measured amorphous phase VO₂ density of 2.81 g cm⁻³ (Supplementary Table 3). The structure was shown to provide an average coordination number of 5.08 for V (with a cutoff for the V-O bond of 2.4 Å), close to the measured value of 4.8¹².

The corresponding references:

12 Yan, P., Zhou, Y., Zhang, B. & Xu, Q. CO₂ Entropy Depletion-induced 2D Amorphous Structure in Non-van der Waals VO₂. *ChemPhysChem*, e202200342 (2022).

The density test results are shown in Supplementary Table 3 (see also Table R1 below).

Table R1. The density test results for amorphous VO₂ powders.

Sample	Volume (cm ³)	Density (g cm ⁻³)	Elapsed Time (mm:ss)	Temperature (°C)
1	0.1802	2.8075	11:13	31.14
2	0.1800	2.8100	13:46	31.18
3	0.1797	2.8141	16:15	31.24
4	0.1804	2.8031	18:49	31.29
5	0.1809	2.7961	21:22	31.29

(Materials Characterization) The density of the amorphous VO₂ powders was measured

by the automatic true density tester (AccuPyc II 1340).

2. Very strong pseudocapacitive charge storage behavior of VO₂-a (Figure S11), making it less likely a candidate for high-capacity cathode materials for K-ion battery.

Response: We appreciate the reviewer for this comment. However, the strong pseudocapacitive charge storage behavior of VO₂ (a) is an important factor for it to be used as a high-capacity potassium cathode material. It should be emphasized that Dunn et al. also wrote a review paper specifically on vanadium-based pseudocapacitive materials for sodium-ion storage¹.

(1) Superiority and classification of pseudocapacitive

Pseudocapacitive materials exhibit battery-like redox reactions that occur at rates comparable to double-layer formation in a capacitive material. Dunn et al. argued that pseudocapacitance provides an opportunity to achieve high energy density at high power density, and an increasing number of studies show that pseudocapacitive materials can fulfill this goal². Fan et al. expressed similar views³. The introduction of pseudocapacitance has become an effective method and has been demonstrated widely in various electrode materials for Li-ion and Na-ion storage. The pseudocapacitive process renders much faster charge transfer than volume lattice diffusion and thus can help to retain the capacity at high current rates. Specifically, pseudocapacitance can be introduced by many methods. For example, nanosizing, increasing electrical conductivity, tuning the crystallinity and crystal phases, introducing mesoporosity, and oxygen deficiency are effective approaches. In particular, it is the optimization of the

Editorial Note: Figure R1 adapted with the permission of Royal Society of Chemistry, from Augustyn, V., Simon, P. & Dunn, B. Pseudocapacitive oxide materials for high-rate electrochemical energy storage. *Energy Environ. Sci.* **7** (2014); permission conveyed through Copyright Clearance Center, Inc.

pseudocapacitive characteristics by phase engineering that is presented in this paper.

There are two main mechanisms of charge storage in pseudocapacitive materials: surface redox pseudocapacitance and intercalation pseudocapacitance. Surface redox pseudocapacitance occurs when ions are electrochemically adsorbed onto the surface or near the surface of a material with a concomitant faradaic charge transfer (Fig. R1a). Intercalation pseudocapacitance occurs when ions intercalate into the tunnels or layers of a redox-active material accompanied by a faradaic charge transfer with no crystallographic phase change (Fig. R1b)⁴. Furthermore, we know that in the battery-type embedding reaction mechanism, the redox process of the electrode material is usually accompanied by a phase transformation. Therefore, the use of materials with a large contribution of pseudocapacitance can effectively mitigate the degradation of electrochemical properties caused by phase transformation.

Fig. R1 Different types of reversible redox mechanisms that give rise to pseudocapacitance: a) redox pseudocapacitance, and b) intercalation pseudocapacitance. Figure adapted from ref. 4.

Cheng et al. made a further generalization. Although both pseudocapacitive storage and battery-type storage inherently undergo redox reactions, a significant distinction is

Editorial Note: Figure R2 adapted with the permission of Royal Society of Chemistry from Gao, Y., Yin, J., Xu, X. & Cheng, Y. Pseudocapacitive storage in cathode materials of aqueous zinc ion batteries toward high power and energy density. *J. Mater. Chem. A* **10**, 9773-9787 (2022); permission conveyed through Copyright Clearance Center, Inc.

that pseudocapacitive storage is not limited by the sluggish reaction kinetics of solid-state diffusion. In the cyclic voltammetry (CV) curves, pseudocapacitive storage exhibits broad oxidation/reduction peaks. On the other hand, a significant separation between the oxidation/reduction peaks of battery-type storage can be observed in the CV curves. Furthermore, in potential (E) versus time (t) plots, battery-type storage exhibits a nearly time-independent potential plateau, whereas the E and t of pseudocapacitance are almost linear, which is similar to that of EDLCs⁵. In our research, the K⁺ ions storage mechanisms of VO₂ (a) consist of both the diffusion and pseudocapacitive controlled redox processes.

Fig. R2 a)-d) CV and e)-h) E-t plots of EDLCs, surface redox pseudocapacitance, intercalation pseudocapacitance and battery-type storage, respectively. Figure adapted from ref. 5.

(2) Pseudocapacitive of vanadium-based materials

In another paper on vanadium-containing materials as cathodes for zinc ion batteries. Golberg et al. concluded that a major advantage of vanadium-based cathodes is that

they are easily able to adopt pseudocapacitive characteristics⁶.

Due to its unique layered structure, V_2O_5 exhibits excellent Zn^{2+} storage capacity. Common optimization strategies, including structural water and ion/molecular pre-embedding, as well as defect engineering, have proven to be effective strategies for improving the pseudocapacitance storage capacity of V_2O_5 ^{7,8}. Vanadium-based sodium cathode ($Fe_5V_{15}O_{39}(OH)_9 \cdot 9H_2O$) has high pseudocapacitance characteristics, resulting in high specific capacity and excellent rate capability⁹. In addition, VO_2 (B) is also considered suitable for fast storage of Li^+ and Na^+ due to its one-dimensional tunnel structure. However, low conductivity and unstable structure still limit its pseudocapacitive storage and rate performance¹⁰.

(3) More pseudocapacitive cathode materials

MXenes are a typical pseudocapacitive energy storage materials that store and transfer charge through rapid oxidation reduction on their surfaces. This energy storage mechanism results in fast reaction kinetics¹¹.

The study of the heterocyclic organic molecule 2,2'-bipyridine-4,4'-dicarboxylic acid (BPDCA) indicated that the intercalation pseudocapacitance with secondary bonding channels of weak hydrogen bonding and interlayer bonding cooperation has the electrochemical performance of high rate and good cyclic stability¹².

The functionalized porphyrin cathode is proposed as a universal cathode material for electrochemical energy storage. The study indicates that the charge storage is mainly controlled by pseudocapacitive contribution, thus achieving ultrafast reaction kinetics, where both cations and anions are stored at the near surface of the porphyrin electrode

through interaction with the nitrogen atoms of the porphyrin complex during the charge and discharge process¹³.

In summary, amorphous VO₂ with a higher pseudocapacitive ratio has higher capacity and better rate performance, which can be considered as an excellent candidate for high-capacity cathode materials for K-ion batteries. This work is also an important extension and supplement to the application of vanadium-based materials with pseudocapacitive in potassium cathode.

Revision: (Page 16) The pseudocapacitive storage mechanism is expected to achieve both high energy and high power density through a fast redox reaction. In addition, when ions are inserted into the tunnels of the active material, intercalation pseudocapacitance occurs with no crystallographic phase change⁵⁹.

The corresponding references:

59 Augustyn, V., Simon, P. & Dunn, B. Pseudocapacitive oxide materials for high-rate electrochemical energy storage. *Energy Environ. Sci.* 7 (2014).

3. The data quality could be better (e.g., XPS, XRD).

Response: We thank the reviewer for pointing this out. In our revised manuscript, we improved the quality of our XPS data. For XRD, there are no obvious peaks for amorphous VO₂ in either the sample grown on carbon fiber cloth or the powder sample, which fully illustrates its amorphous character. Moreover, due to the influence of carbon fiber cloth, the peaks of the crystalline phase of VO₂ may be a little weak, but both the main peak (110) of B phase VO₂ at 25.3° and the main peak (011) of M phase VO₂ at 27.8° are obvious, and the rest of the peaks correspond with those in the JCPDS card

one by one. Nevertheless, we have optimized the XRD data as much as possible.

Revision: The updated XPS results are shown in Supplementary Fig. 9 (see also Fig. R3 below).

Fig. R3 XPS spectra of V 2p of a) VO₂ (a), b) VO₂ (B) and c) VO₂ (M) at pristine state and discharged state of 1 V.

The revised XRD results are shown in Supplementary Fig. 1 and 6 (see also Fig. R4 and 5 below).

Fig. R4 XRD patterns and the corresponding JCPDS data of VO₂ (a), VO₂ (B), and VO₂ (M).

Fig. R5 XRD patterns of VO₂ (a) powder, VO₂ (B) powder, and VO₂ (M) powder. The peaks marked as “▼” come from the V₈O₁₅ impurity.

- 1 Wei, Q., DeBlock, R. H., Butts, D. M., Choi, C. & Dunn, B. Pseudocapacitive Vanadium-based Materials toward High-Rate Sodium-Ion Storage. *Energy Environ. Mater.* **3**, 221-234 (2020).
- 2 Choi, C. *et al.* Achieving high energy density and high power density with pseudocapacitive materials. *Nat. Rev. Mater.* **5**, 5-19 (2019).
- 3 Chao, D. & Fan, H. J. Intercalation Pseudocapacitive Behavior Powers Aqueous Batteries. *Chem* **5**, 1359-1361 (2019).
- 4 Augustyn, V., Simon, P. & Dunn, B. Pseudocapacitive oxide materials for high-rate electrochemical energy storage. *Energy Environ. Sci.* **7** (2014).
- 5 Gao, Y., Yin, J., Xu, X. & Cheng, Y. Pseudocapacitive storage in cathode materials of aqueous zinc ion batteries toward high power and energy density. *J. Mater. Chem. A* **10**, 9773-9787 (2022).
- 6 Lewis, C. E. M. *et al.* Vanadium-containing layered materials as high-performance cathodes for aqueous zinc-ion batteries. *Adv. Mater. Technol.* **7**, 2100505 (2021).
- 7 Yan, M. *et al.* Water-Lubricated Intercalation in V₂O₅·nH₂O for High-Capacity and High-Rate Aqueous Rechargeable Zinc Batteries. *Adv Mater* **30** (2018).
- 8 Liu, S. *et al.* Tuning the Kinetics of Zinc-Ion Insertion/Extraction in V₂O₅ by In Situ Polyaniline Intercalation Enables Improved Aqueous Zinc-Ion Storage Performance. *Adv Mater* **32**, e2001113 (2020).
- 9 Wei, Q. *et al.* Pseudocapacitive layered iron vanadate nanosheets cathode for ultrahigh-rate lithium ion storage. *Nano Energy* **47**, 294-300 (2018).
- 10 Chao, D. *et al.* Graphene quantum dots coated VO₂ arrays for highly durable electrodes for Li and Na ion batteries. *Nano Lett.* **15**, 565-573 (2015).
- 11 Li, X. *et al.* Intrinsic voltage plateau of a Nb₂CT_x MXene cathode in an aqueous electrolyte induced by high-voltage scanning. *Joule* **5**, 2993-3005 (2021).
- 12 Hu, Z. *et al.* Secondary Bonding Channel Design Induces Intercalation Pseudocapacitance toward Ultrahigh-Capacity and High-Rate Organic Electrodes. *Adv Mater* **33**, e2104039 (2021).
- 13 Feng, X. *et al.* A bipolar organic molecule toward a universal pseudocapacitive cathode

for stable dual ion charge storage. *Energy Storage Mater.* **42**, 454-463 (2021).

We sincerely hope our detailed explanations are satisfactory, and this reviewer will recommend *Nature Communications* to accept our revised manuscript.

Reviewer #2 (Remarks to the Author):

The manuscript presents a combined experimental-computational study of the use of VO₂ as a potassium ion battery cathode. The key finding is that amorphization facilitates K storage. Vanadia phases continue to be promising for beyond-Li batteries and as vanadia present rich phase diagrams, phase selection is an important issue. Therefore this is in principle impactful and publishable if the work is done right. The following issues need to be addressed to ensure that:

Response: We sincerely thank the reviewer for the positive comments. We strongly agree that vanadia phases are promising for beyond-Li batteries, and vanadia phase selection is an important issue. Therefore, phase engineering is a research direction of great significance. Thank you for your professional comments, which have greatly improved our article. Accordingly, we have recalculated all the DFT parts in the article. Below, we answer your questions one by one.

1- The authors present a combined experimental-DFT study but refer only to experimental prior work. There is a relatively long line of computational research arguing for the benefits of amorphization for the storage of large(er than Li) cations where crystalline phases do not work well. This was rationalized earlier in DFT studies on Si and titania in J. Phys. Chem. C 119, 13496–13501 (2015), J. Power Sources, 278, 197-202 (2015), Comput. Mater. Sci., 94, 214-217 (2014) which were later borne out experimentally. Specifically promise of amorphization for K storage was shown in MRS Advances 1, 3069–3074 (2016) on the example of carbon. Regarding relative

Editorial Note: Figure R6 adapted with the permission of Royal Society of Chemistry, from Kulish, Vadym V. & Manzhos, S. Comparison of Li, Na, Mg and Al-ion insertion in vanadium pentoxides and vanadium dioxides. *RSC Adv.* **7**, 18643-18649 (2017); permission conveyed through Copyright Clearance Center, Inc.

performance of different vanadia phases, a comprehensive review of vanadia phases for use in metal ion batteries was published in *J. Phys. D* **3**, 083001 (2020) that conceptualizes many of the issues the authors are dealing with. The present paper would be well served if it contextualized itself better in the light of those prior research trends that it continues and which solidified the case for using amorphous phases.

Response: Thank you for your professional insight. We did some literature review before our calculations for VO₂. For example, we referred to the article in *RSC Adv.*, 2017, 7, 18643–18649 for the selection of B phase VO₂ insertion sites¹.

Fig. R6 VO₂ (B) supercell with possible insertion sites. Figure adapted from ref. 1.

We also referred to the article in *MRS Communications*, 2017, 7(4): 819-825 for the migration energy and the DOS of B phase VO₂².

[REDACTED]

Fig. R7 a) Barriers for the diffusion of K in K_{0.03}VO₂ (B), α-K_{0.06}V₂O₅, β-K_{0.01}V₂O₅ between two equivalent lowest energy insertion sites. b) Spin-polarized density of states of VO₂ (B). Figure adapted from ref. 2.

In addition, we also referred to the relevant calculation of M-phase VO₂ in *Phys. Chem. Chem. Phys.*, 2017, 19, 22538-22545³. However, these are obviously not sufficient. In

the original writing, we paid more attention to the experimental results and characterization, but neglected the references to computational aspects. Especially the theoretical calculation of amorphous materials is complicated, which is a big mistake on our part. The papers you have listed are very relevant to our research, and we have studied them carefully and cited them in our revised manuscript. This will provide the readers with a more comprehensive background of computational studies and a sufficient argument for the benefits of amorphization for alkali metal storage.

Revision: (Introduction) A few studies have focused on amorphization for cation storage and explained the advantages of the amorphous phase from the perspective of theoretical calculations¹⁴⁻¹⁸. The superiority of using amorphous phases in energy storage applications can be further verified and solidified if more convincing experimental results can be combined with theoretical calculations in the context of continuing previous research trends.

The corresponding references:

- 14 Legrain, F. et al. Amorphous (Glassy) Carbon, a Promising Material for Sodium Ion Battery Anodes: a Combined First-Principles and Experimental Study. *J. Phys. Chem. C* 119, 13496-13501 (2015).
- 15 Legrain, F., Malyi, O. & Manzhos, S. Insertion energetics of lithium, sodium, and magnesium in crystalline and amorphous titanium dioxide: A comparative first-principles study. *J. Power Sources* 278, 197-202 (2015).
- 16 Legrain, F., Malyi, O. I. & Manzhos, S. Comparative computational study of the energetics of Li, Na, and Mg storage in amorphous and crystalline silicon. *Comput. Mater. Sci* 94, 214-217 (2014).
- 17 Legrain, F., Kotsis, K. & Manzhos, S. Mg and K Insertion in Glassy Amorphous Carbon vs Graphite as Potential Anode Materials: an Ab Initio Study. *MRS Adv.* 1, 3069-3074 (2016).
- 18 Koch, D. & Manzhos, S. Ab initio modeling and design of vanadia-based electrode materials for post-lithium batteries. *J. Phys. D: Appl. Phys.* 53 (2020).

2- The authors appear to report the lowest diffusion barrier in the amorphous phase, however, this does not guarantee that long-range diffusion is more favored vs a crystal, as in the a-VO₂ there will be multiple non-equivalent diffusion barriers. One needs to show a long-range path (through the entire simulation cell)

Response: Thank you very much for this suggestion. Accordingly, we have recalculated different diffusion paths in the amorphous phase VO₂. We have chosen three long-range paths from different directions through the entire simulation cell. Their migration energy barriers are all lower than those of the crystalline phase, which fully illustrates the superiority of the amorphous phase.

Revision: (Page 14) Multiple non-equivalent diffusion paths for K⁺ ions are present in the amorphous configuration. To ensure that the amorphous is more favorable than the crystalline phase, three long-range paths from different directions through the entire simulation cell were selected (Fig. 4a-c). The preferred diffusion directions in the crystalline structure were selected by the largest channel size present in the structure (Fig. 4d-e), and the relative migration energies are shown in Fig. 4f.

(Page 14) The migration energies of all three paths of the amorphous phase are relatively small, which are 2.15, 2.74, and 1.85 eV, respectively. A diffusion barrier of 3.54 eV is found in VO₂ (B), for which the pathway for K⁺ diffusion is mainly along the *b*-axis, as reported previously⁵⁷. The typical diffusion pathway for K⁺ transport in VO₂ (M) is within the open tunnel³⁵, and since the K⁺ ion has no stable insertion site, its diffusion barrier is calculated for reference only.

The corresponding references:

- 57 Koch, D., Kulish, V. V. & Manzhos, S. A first-principles study of potassium insertion in crystalline vanadium oxide phases as possible potassium-ion battery cathode materials. *MRS Commun.* 7, 819-825 (2017).
- 35 Kulish, V. V., Koch, D. & Manzhos, S. Ab initio study of Li, Mg and Al insertion into rutile VO₂: fast diffusion and enhanced voltages for multivalent batteries. *Phys Chem Chem Phys* 19, 22538-22545 (2017).

The updated diffusion barrier results are shown in Fig. 4a-f (see also Fig. R8a-f below).

Fig. R8 DFT calculations of K storage. The K migration paths of the a) - c) VO₂ (a), d) VO₂ (B), and e) VO₂ (M). f) Migration energy of K⁺ ions for VO₂ in the three different phases. The DOS before and after K⁺ insertion for g) VO₂ (a), h) VO₂ (B), and i) VO₂ (M). The partial densities of states for the three atomic species are indicated in yellow (vanadium), green (oxygen), and purple (potassium, magnified by a factor of 200 for better visibility).

3- Why was Grimme D3 used?? These are not layered phases, so dispersion corrections are not needed and will in fact make the accuracy worse. Moreover, the Grimme correction should not be applied to K as it is fully ionized (= no valence shell to participate in dispersion interactions). This issue could throw off the insertion energies by tenths of an eV.

Response: We agree that the issue you mentioned is significant for the accuracy of the calculation. It was not reasonable for us to use Grimme D3 before. In the revised manuscript, we did not use dispersion corrections and recalculated all the contents.

Revision: (Computational methods part of Supplementary information) The sentence “Grimme’s DFT-D3 methodology was used to describe the dispersion interactions.” has been deleted.

4- the parameters of the generated amorphous structure should be given and compared to experimental a-VO₂ (density, energy metastability, coordination number statistics)

Response: Thank you very much for this comment. To ensure the accuracy and representativeness of the established amorphous configurations, the parameters of it should agree with the experimentally generated structures. The density and coordination numbers of the amorphous structures obtained from the calculations are compared with the experimental VO₂ (a), and the results were in good agreement.

Revision: (Computational methods part of Supplementary information) The initial density is about 2.8 g cm⁻³, which agrees with the measured amorphous phase VO₂ density of 2.81 g cm⁻³ (Supplementary Table 3). The structure was shown to provide an average coordination number of 5.08 for V (with a cutoff for the V-O bond of 2.4 Å), close to the measured value of 4.8¹². The amorphous structure is less stable than the crystalline phase by 0.22 eV per atom.

The corresponding references:

12 Yan, P., Zhou, Y., Zhang, B. & Xu, Q. CO₂ Entropy Depletion-induced 2D Amorphous Structure in Non-van der Waals VO₂. *ChemPhysChem*, e202200342

(2022).

The density test results are shown in Supplementary Table 3 (see also Table R1 below).

Table R1. The density test results for amorphous VO₂ powders.

Sample	Volume (cm ³)	Density (g cm ⁻³)	Elapsed Time (mm:ss)	Temperature (°C)
1	0.1802	2.8075	11:13	31.14
2	0.1800	2.8100	13:46	31.18
3	0.1797	2.8141	16:15	31.24
4	0.1804	2.8031	18:49	31.29
5	0.1809	2.7961	21:22	31.29

(Materials Characterization) The density of the amorphous VO₂ powders was measured by the automatic true density tester (AccuPyc II 1340).

5- the sentence "The conjugated gradient method was applied with a smearing width of 0.15 eV" does not make much sense. Do the authors mean smearing of electronic occupancies?

Response: We are very sorry for the confusion caused by our original statement. What we mean is smearing of electronic occupancies. However, in fact, we have also described it elsewhere as "using the Gaussian smearing method and a width of 0.05 eV". We have revised the sentence.

Revision: (Computational methods part of Supplementary information) The conjugated gradient method was applied, and the convergence criteria for the energy and force were 10^{-4} eV/cell and 0.05 eV/Å, respectively.

6- Figs 1 and 3 are too crowded. It is better to split the exp and the models into different figures.

Response: Thank you for this suggestion. Accordingly, we changed the original Fig. 1 to the new Fig. 1 and Fig. 2 in the revised manuscript. We also modified the original Fig. 3 by moving the experimental part to the supplementary information.

Revision: The new Fig. 1 (see also Fig. R9 below).

Fig. R9 Investigation of potassium storage capability of VO₂ in different phases. a) HAADF-STEM image of VO₂ (M) with corresponding elemental mapping of b) V, c) O, and d) K. e) and f) The HRTEM images of the tan and dark blue squared areas in Fig. 1a. Inset: The corresponding FFT patterns. g) The simulated diffraction pattern of VO₂ (M) along the [010] zone axis. Schematic illustrations for h) pristine VO₂ (a) and K-embedded VO₂ (a). i) Crystalline VO₂ (M). The blue, yellow, and purple balls correspond to V, O, and K atoms, respectively.

The new Fig. 2 and Fig. 4 (see also Fig. R10 and R8).

7- the authors refer to the same K positions K1, K2, K3 in the crystalline and amorphous case. But in a-VO₂ there will be multiple and different insertion sites, certainly more than 3 (if the amorphous structure is realistic). Insertion energies in a representative set of such sites should be given.

Response: Thank you for this constructive suggestion. We have selected a large number of embedded potassium sites in the amorphous configuration, performed binding energy calculations, and put the results in the new Figure 2. We initially selected 49 potassium storage sites in the amorphous configuration according to the principle of uniform distribution, and their distances from V and O atoms were farther than 1.45 Å. After structural optimization, there are 25 sites with different locations remaining. In this way, we performed the binding energy statistics versus the bulk metal reference state for K insertion.

Revision: (Page 7) To identify potential insertion sites in VO₂ (a), 49 potassium storage sites were selected in the amorphous configuration according to the principle of uniform distribution, and their distances from V and O atoms were farther than 1.45 Å. After structural optimization, 25 sites with different positions remained. In Figure 2a-e, the insertion sites of VO₂ (a) are shown in groups of five positions from low to high binding energy. Three non-equivalent insertion sites are present for both VO₂ (B) and VO₂ (M) (Fig. 2f and g), which differ by coordination number and bond distances^{34,35}. Multiple different insertion sites of K⁺ ions result in varying binding energies, as depicted in Fig. 2h.

(Page 7) The only stable is the eightfold coordinated insertion site for VO₂ (B).

However, it is still higher than most amorphous sites.

The corresponding references:

- 34 Kulish, Vadym V. & Manzhos, S. Comparison of Li, Na, Mg and Al-ion insertion in vanadium pentoxides and vanadium dioxides. *RSC Adv.* 7, 18643-18649 (2017).
- 35 Kulish, V. V., Koch, D. & Manzhos, S. Ab initio study of Li, Mg and Al insertion into rutile VO₂: fast diffusion and enhanced voltages for multivalent batteries. *Phys Chem Chem Phys* 19, 22538-22545 (2017).

The revised K⁺ ions insertion sites and the corresponding binding energies are shown in Fig. 2 (see also Fig. R10 below).

Fig. R10 The K⁺ ions insertion sites in a) - e) VO₂ (a), f) VO₂ (B), and g) VO₂ (M). h)

The binding energies of K⁺ ions at different positions in VO₂ (a), VO₂ (B), and VO₂ (M).

8- something appears to be not right with the PDOS. At that K concentration are those? PDOS should be shown before and after K insertion. The semiconductor phases such as M should show a clear non-polarized DOS with a gap, which upon insertion of one K into the simulation cell should show metallic character with spin polarization with

either a conduction band delocalized state (without +U) or a localized gap state (with +U which should be applied in this case but apparently is not). What is shown is something else.

Response: Thank you for your careful review of our work. We admit that there were some problems with our previous DOS results. In the revised manuscript, we present the DOS of different phases of VO₂ before and after K insertion as per your suggestion.

Revision: (Page 15) Specifically, the pristine VO₂ (a) has a spin-polarized metallic electronic structure with no gap around the Fermi level in the DOS (Fig. 4g). The pristine VO₂ (B) is a spin-polarized semi-metal with a band gap of 0.29 eV (Fig. 4h), which is similar to previous reports^{18,57}. It can be found that the K insertion does not noticeably affect the electronic structure of the VO₂ host both in VO₂ (a) and VO₂ (B). In contrast, the semiconductor phase VO₂ (M) shows a clear non-polarized DOS with a gap of 0.99 eV, which upon insertion of one K into the simulation cell, shows a smaller gap with spin polarization with a localized gap state (Fig. 4i).

The corresponding references:

- 18 Koch, D. & Manzhos, S. Ab initio modeling and design of vanadia-based electrode materials for post-lithium batteries. *J. Phys. D: Appl. Phys.* 53 (2020).
- 57 Koch, D., Kulish, V. V. & Manzhos, S. A first-principles study of potassium insertion in crystalline vanadium oxide phases as possible potassium-ion battery cathode materials. *MRS Commun.* 7, 819-825 (2017).

(Computational methods part of Supplementary information) A rotationally invariant Hubbard correction was applied, as proposed by Dudarev et al., with (U-J) = 3.1 eV for V d states⁹.

The corresponding references:

9 Koch, D. & Manzhos, S. First-Principles Study of the Calcium Insertion in Layered and Non-Layered Phases of Vanadia. *MRS Adv.* **3**, 3507-3512 (2018).

The updated DOS results are shown in Fig. 4g-i (see also Fig. R8g-i below).

Fig. R8 DFT calculations of K storage. The K migration paths of the a) - c) VO₂ (a), d) VO₂ (B), and e) VO₂ (M). f) Migration energy of K⁺ ions for VO₂ in the three different phases. The DOS before and after K⁺ insertion for g) VO₂ (a), h) VO₂ (B), and i) VO₂ (M). The partial densities of states for the three atomic species are indicated in yellow (vanadium), green (oxygen), and purple (potassium, magnified by a factor of 200 for better visibility).

The above suggests the need for a major revision.

- 1 Kulish, Vadym V. & Manzhos, S. Comparison of Li, Na, Mg and Al-ion insertion in vanadium pentoxides and vanadium dioxides. *RSC Adv.* **7**, 18643-18649 (2017).
- 2 Koch, D., Kulish, V. V. & Manzhos, S. A first-principles study of potassium insertion in crystalline vanadium oxide phases as possible potassium-ion battery cathode materials. *MRS Commun.* **7**, 819-825 (2017).
- 3 Kulish, V. V., Koch, D. & Manzhos, S. Ab initio study of Li, Mg and Al insertion into rutile VO₂: fast diffusion and enhanced voltages for multivalent batteries. *Phys Chem Chem Phys* **19**, 22538-22545 (2017).

We sincerely appreciate the constructive feedback, which has vastly improved the

accuracy of DFT calculations and, thus, the overall quality of the revised manuscript.

We hope our detailed explanations are satisfactory, and this reviewer will recommend

Nature Communications to accept our revised manuscript.

Reviewer #3 (Remarks to the Author):

In this manuscript, the application of phase engineering has been realized by constructing an amorphous phase VO₂ cathode, and the surprising effects have been achieved in potassium ion batteries. The idea of this paper is clear, especially the use of a large number of TEM characterizations provides a good confirmation of the effect of the phase engineering proposed by the authors. Compared with the crystalline phase, the amorphous phase VO₂ not only has stable potassium storage sites, but also shows little deformation during battery use. The analysis of the HRTEM is exact, such as the analysis of the lattice, the diffraction pattern, and the zone axis. This study establishes a relationship between phase and electrochemical properties, which is of great importance. Hence, I would be glad to recommend the acceptance of this paper in Nature Communications after addressing the following minor issues.

Reply: We sincerely thank the reviewer for appreciating the contents of our work and recommending it for publication in the Nature Communications. Below, we have addressed the reviewer's comments point to point.

(1) Please explain in detail how the capacity of pure carbon cloth is measured in Supplementary Figure 8.

Response: Thank you for this question. More details are needed in our statement about the cycle test of pure carbon fiber cloth (CF). First, the capacity was calculated based on the actual mass of the pure CF itself. Then, the current during the cycle test was

empirically based on the other electrode with active materials. Assuming 0.8 mg of active material on pure CF, and to measure 50 mA g^{-1} current density during cycling, its current should be set to 0.04 mA. We have added the experimental details in the revised manuscript.

Revision: (Electrochemical measurements) When measuring the cycle of pure CF, the capacity was calculated based on the actual mass itself, and the current was obtained empirically from the other electrodes with active materials, which was about 0.04 mA.

(2) Additional experimental details on the full battery assembly should be provided, such as the proportion of each component on the negative electrode and the amount of electrolyte injection.

Response: Thanks for this comment. We are very sorry to ignore the description of the proportion of the negative electrode and the amount of electrolyte injection for the full cell. We have added the related details in the revised manuscript.

Revision: (Electrochemical measurements) whose slurry was prepared by 80 wt% active material, 10 wt% conductive carbon black (Super P), and 10 wt% CMC onto an Al foil.

(Electrochemical measurements) and its injection volume was about 0.4 ml.

(3) Amorphous VO_2 has no characteristic peak in XRD characterization. In this paper, the amorphous sample is identified as VO_2 based only on the XPS results. Additional evidence is needed, such as the results on the ratio of elements in the EDS.

Response: We appreciate the reviewer’s valuable suggestion. We totally understand the reviewer’s concern and agree that the EDS data would be a good characterization index for the correct identification of the amorphous sample as VO₂. As suggested, in the revised manuscript, we included the EDS characterization to get the ratio of elements.

Revision: (Page 9) In particular, the energy dispersive spectrometer (EDS) results of VO₂ (a) state in Supplementary Fig. 3 confirm the existence of V and O, and their ratios are very close to the chemical formula atomic ratio, as shown in Supplementary Table 1.

The EDS result is shown in Supplementary Fig. 3 (see also Fig. R11 below).

Fig. R11 EDS results of VO₂ (a).

The atomic ratio are shown in Supplementary Table 1 (see also Table R2 below).

Table R2. The atomic ratio of elements from EDS.

Element	Wt %	Atomic %
C	76.09	87.92
O	9.37	8.12
V	14.54	3.96

(4) The authors should carefully check the main text to ensure no errors. For example, “The DFT calculations likewise confirmed that...” should be “The DFT calculations likewise confirmed that...”. (In Discussion)

Reply: We thank the reviewer for this comment and are sorry for the previous errors.

We have double-checked the manuscript and tried our best to correct the errors. Few examples are listed below.

Revision: (Page 5) Change “The high-angle annular dark-field (HAADF)” to “The high-angle annular dark-field (HAADF)”

(Page 5) Change “Figure 1h is schematically illustrates both...” to “Figure 1h schematically illustrates both...”.

(Page 8) Change “It indicates this VO₂ is amorphous.” to “It indicates that this VO₂ is amorphous.”

(Page 22) Change “After the battery was bent 1000 times, the LEDs light is still glow (Fig. 6h)” to “After the battery is bent 1000 times, the LEDs light still glows (Fig. 6h)”.

(In Discussion) Change “The DFT calculations likewise confirmed that...” to “The DFT calculations likewise confirmed that...”.

(Materials synthesis) Change “..., except without adding the CF.” to “..., but without adding the CF.”

(5) Authors need to read more articles about phase engineering carefully and cite them selectively. Do the results of this paper have a generality to lithium ion batteries?

Reply: Thanks to your suggestion, we have read as much literature as possible on phase engineering as well as amorphous phases, and selected most relevant articles and cited them, in our manuscript.

Phase has emerged as an important structural parameter in addition to composition,

morphology, architecture, facet, size, and dimensionality that determines the properties and functionalities of nanomaterials^{1,2}. Therefore, in recent years, phase engineering has become an effective strategy to modulate the physicochemical properties of nanomaterials and promote their applications in various important fields, such as energy storage^{3,4}, conversion^{5,6}, and catalytic reactions^{7,8}. The unconventional phase materials often have unique properties⁹, and the amorphous phase is one of them.

In our work, phase engineering mainly describes the advantages of the amorphous phase over the crystalline phase for energy storage applications. Amorphous materials exhibit random atomic arrangement or short-range order over just a few atoms, resulting in distorted lattices and dangling bonds. The rich defects and unsaturated coordination sites can provide more active sites for alkali-ion storage. In addition, the amorphous phase can mitigate lattice strains induced by ions insertion/extraction and thus prevent the mechanical failure of the electrode.

There have been some studies on amorphization for the storage of Li^{10,11}, Na¹², Mg¹³, and K¹⁴, explaining the advantages of the amorphous phase from the perspective of theoretical calculation. To provide the readers with a more comprehensive background of computational studies and a sufficient argument for the benefits of amorphization for alkali metal storage, most relevant literature is cited in our article. This is a good indication that phase engineering, especially the application of amorphous phase, is a general tool, whether for lithium, sodium, or other energy storage systems such as magnesium.

Revision: (Introduction) **Consequently, phase engineering directly determines the**

properties and functioning of nanomaterials^{8,9}. The unconventional phase materials often have unique properties¹⁰, and the amorphous phase is one of them.

(Introduction) A few studies have focused on amorphization for cation storage and explained the advantages of the amorphous phase from the perspective of theoretical calculations¹⁴⁻¹⁸. The superiority of using amorphous phases in energy storage applications can be further verified and solidified if more convincing experimental results can be combined with theoretical calculations in the context of continuing previous research trends.

The corresponding references:

- 8 Chen, Y. et al. Phase engineering of nanomaterials. *Nat. Rev. Chem.* 4, 243-256 (2020).
- 9 Li, H. et al. Phase Engineering of Nanomaterials for Clean Energy and Catalytic Applications. *Adv. Energy Mater.* 10 (2020).
- 10 Liu, J., Huang, J., Niu, W., Tan, C. & Zhang, H. Unconventional-Phase Crystalline Materials Constructed from Multiscale Building Blocks. *Chem Rev* 121, 5830-5888 (2021).
- 14 Legrain, F. et al. Amorphous (Glassy) Carbon, a Promising Material for Sodium Ion Battery Anodes: a Combined First-Principles and Experimental Study. *J. Phys. Chem. C* 119, 13496-13501 (2015).
- 15 Legrain, F., Malyi, O. & Manzhos, S. Insertion energetics of lithium, sodium, and magnesium in crystalline and amorphous titanium dioxide: A comparative first-principles study. *J. Power Sources* 278, 197-202 (2015).
- 16 Legrain, F., Malyi, O. I. & Manzhos, S. Comparative computational study of the energetics of Li, Na, and Mg storage in amorphous and crystalline silicon. *Comput. Mater. Sci* 94, 214-217 (2014).
- 17 Legrain, F., Kotsis, K. & Manzhos, S. Mg and K Insertion in Glassy Amorphous Carbon vs Graphite as Potential Anode Materials: an Ab Initio Study. *MRS Adv.* 1, 3069-3074 (2016).
- 18 Koch, D. & Manzhos, S. Ab initio modeling and design of vanadia-based electrode materials for post-lithium batteries. *J. Phys. D: Appl. Phys.* 53 (2020).

1 Chen, Y. et al. Phase engineering of nanomaterials. *Nat. Rev. Chem.* 4, 243-256 (2020).

2 Li, H. et al. Phase Engineering of Nanomaterials for Clean Energy and Catalytic Applications. *Adv. Energy Mater.* 10 (2020).

- 3 Geng, X. *et al.* Two-Dimensional Water-Coupled Metallic MoS₂ with Nanochannels for
Ultrafast Supercapacitors. *Nano Lett* **17**, 1825-1832 (2017).
- 4 Acerce, M., Voiry, D. & Chhowalla, M. Metallic 1T phase MoS₂ nanosheets as
supercapacitor electrode materials. *Nat. Nanotechnol.* **10**, 313-318 (2015).
- 5 Li, Q. *et al.* Pressure-Induced Phase Engineering of Gold Nanostructures. *J Am Chem
Soc* **140**, 15783-15790 (2018).
- 6 Lu, S. *et al.* Crystal Phase Control of Gold Nanomaterials by Wet-Chemical Synthesis.
Acc Chem Res **53**, 2106-2118 (2020).
- 7 Yin, Y. *et al.* Synergistic Phase and Disorder Engineering in 1T-MoSe₂ Nanosheets for
Enhanced Hydrogen-Evolution Reaction. *Adv Mater* **29** (2017).
- 8 Ge, Y. *et al.* Preparation of fcc-2H-fcc Heterophase Pd@Ir Nanostructures for High-
Performance Electrochemical Hydrogen Evolution. *Adv Mater* **34**, e2107399 (2022).
- 9 Liu, J., Huang, J., Niu, W., Tan, C. & Zhang, H. Unconventional-Phase Crystalline
Materials Constructed from Multiscale Building Blocks. *Chem Rev* **121**, 5830-5888
(2021).
- 10 Legrain, F., Malyi, O. & Manzhos, S. Insertion energetics of lithium, sodium, and
magnesium in crystalline and amorphous titanium dioxide: A comparative first-
principles study. *J. Power Sources* **278**, 197-202 (2015).
- 11 Koch, D. & Manzhos, S. Ab initio modeling and design of vanadia-based electrode
materials for post-lithium batteries. *J. Phys. D: Appl. Phys.* **53** (2020).
- 12 Legrain, F. *et al.* Amorphous (Glassy) Carbon, a Promising Material for Sodium Ion
Battery Anodes: a Combined First-Principles and Experimental Study. *J. Phys. Chem. C*
119, 13496-13501 (2015).
- 13 Legrain, F., Malyi, O. I. & Manzhos, S. Comparative computational study of the
energetics of Li, Na, and Mg storage in amorphous and crystalline silicon. *Comput.
Mater. Sci* **94**, 214-217 (2014).
- 14 Legrain, F., Kotsis, K. & Manzhos, S. Mg and K Insertion in Glassy Amorphous Carbon vs
Graphite as Potential Anode Materials: an Ab Initio Study. *MRS Adv.* **1**, 3069-3074
(2016).

We sincerely appreciate this reviewer's constructive feedback, which has helped improve the overall quality of the revised manuscript. We hope our detailed explanations are satisfactory, and this reviewer will recommend *Nature Communications* to accept our revised manuscript.

Reviewer #4 (Remarks to the Author):

Phase engineering is an effective strategy to regulate the properties, functions, and applications of nanomaterials by rationally tuning their atomic arrangements. In this work, at the atomic level, the authors found that the crystalline region in the M phase VO₂ is not able to store potassium, while the residual amorphous region remaining from the annealing preparation has a stable potassium intercalation ability, which is an interesting phenomenon. The authors first realized the application of VO₂ as a PIBs cathode by using an amorphous phase, demonstrating the effectiveness of phase engineering. Moreover, the amorphous VO₂ shows an excellent sustained cycling performance of 8500 cycles. Overall, this is a meaningful study that sheds light on the exploration of high-performance rechargeable batteries and is suitable for publication in Nature Communications. The authors need to make the following revisions before publication.

Reply: We thank the reviewer for endorsing our work, and below, we carefully replied to the questions one by one and revised the manuscript based on these comments.

1. The direction of the coordinate axes is not given in Fig. 1h, is it shared with Fig. 1i?

Reply: We thank this reviewer for this useful suggestion. The direction of the coordinate axes in Fig. 1h is inconsistent with Fig. 1i. We have made corresponding revisions in the revised manuscript.

Revision: The new Fig. 1 is shown in revised manuscript (see also Fig. R9 below).

Fig. R9 Investigation of potassium storage capability of VO₂ in different phases. a) HAADF-STEM image of VO₂ (M) with corresponding elemental mapping of b) V, c) O, and d) K. e) and f) The HRTEM images of the tan and dark blue squared areas in Fig. 1a. Inset: The corresponding FFT patterns. g) The simulated diffraction pattern of VO₂ (M) along the [010] zone axis. Schematic illustrations for h) pristine VO₂ (a) and K-embedded VO₂ (a). i) Crystalline VO₂ (M). The blue, yellow, and purple balls correspond to V, O, and K atoms, respectively.

2. Please include the method and description of the simulated diffraction pattern in the supplementary information.

Reply: Thank you for this comment. We have added the method and description of the simulated diffraction pattern to the Supplementary information.

Revision: (Detailed of the simulated diffraction pattern of Supplementary information)

The simulated diffraction pattern was performed with the tool cellViewer in CrysTBox software. This tool can show the material from different perspectives in the direct

atomic lattice (cell view) and reciprocal lattice (diffraction view). The only required input of this tool is the CIF of the sample material. In the theoretical spot diffraction pattern, individual diffraction spots are represented by disks. The darker the spot appears, the better satisfied the diffraction condition is. Crosses represent extremely weak or forbidden reflections⁴.

The corresponding references:

- 4 Klinger, M. & Jager, A. Crystallographic Tool Box (CrysTBox): automated tools for transmission electron microscopists and crystallographers. *J. Appl. Crystallogr.* 48, 2012-2018 (2015).

3. Fig. 5g and i are suggested to use the double abscissa, the lower x-axis is the capacity per unit mass, and the upper x-axis is the real capacity of the full battery.

Reply: Thanks for this excellent suggestion. We have added the real capacity in new Fig. 6g and i following your suggestion.

Revision: The revised data are shown in new Fig. 6g and i (see also Fig. R12g and i below).

Fig. R12 The electrochemical performance of the full battery. a) Schematic diagram of the K ion full battery. b) An optical photograph of the cross-section of the soft-pack battery. c) Discharge/charge profiles of the half cells and the full battery. d) Rate and e) cycling performance at 200 mA g^{-1} for $\text{VO}_2(\text{a})/\text{nano-graphite}$ in full battery. Image of LEDs powered by a soft-pack battery composed of $\text{VO}_2(\text{a})$ after bending f) once and h) 1000 times. Discharge/charge profiles of the second cycle after bending g) once and i) 1000 times compared to the initial.

4. In Fig. 5f and h, the author can try to drive something with greater power, such as a light plate composed of multiple LEDs, which is more convincing than lighting such a single small bulb.

Reply: Thank you for this valuable suggestion. Accordingly, we have successfully powered multiple LEDs.

Revision: The revised pictures are shown in new Fig. 6f and h (see also Fig. R12f and

h).

5. It is suggested to supplement the SEM morphology characterization of the flexible electrode after multiple bending, and compare it with the initial state.

Reply: Your suggestion is very useful. We have performed SEM characterization of the flexible VO₂ (a) electrode after bending it 1000 times. It can be seen that multiple bending does not have a significant effect on the morphology of the flexible electrode.

Revision: (Page 22) Furthermore, the FE-SEM images of the flexible electrode after multiple bends (Supplementary Fig. 22) also show essentially the same morphology as the initial state.

The additional FE-SEM images are shown in Supplementary Fig. 22 (see also Fig. R13 below).

Fig R13. The FE-SEM images of VO₂ (a) after bending 1000 times at charged states.

6. The full name of the XPS analysis is not provided anywhere, only the abbreviation. Please add the full name of the characterization upon the first usage of the abbreviation.

Reply: Thank you for pointing out this oversight. We have made the changes in the revised manuscript.

Revision: (Page 12) The K storage mechanism of VO₂ was investigated via *ex situ* X-ray photoelectron spectroscopy (XPS) in Supplementary Fig. 9.

We greatly appreciate the constructive feedback, which has vastly improved the overall quality of the revised manuscript. We hope our detailed explanations are satisfactory, and this reviewer will recommend that *Nature Communications* accept our revised manuscript.

REVIEWER COMMENTS

Reviewer #3 (Remarks to the Author):

The responses to the reviewers' comments are appropriate and the revised version is acceptable for publication in my opinion

Reviewer #4 (Remarks to the Author):

The authors have addressed all my concerned issues. However, I suggest that the authors had better polish the language before the acceptance.

For the comments from Reviewer #1 and #2:

After reading the authors' response to reviewers, it looks that they have addressed the questions raised by the reviewers.

The reviewer 1's main concern is that "the construction of the simulation model and the determination of the amorphous phase is very challenging." And also he/she thought that "the material with strong pseudocapacitive storage behavior makes it less likely a candidate for high-capacity cathode materials for K-ion batteries." The reviewer 2 was also mainly concerned about computational aspect. According to the reviewers' valuable comments, the authors have supplemented the relevant evidences, which considerably improved the quality of this work.

However, before accepting, the authors should further polish the logics and language of this manuscript. In particular, the author should further polish the abstract, introduction and conclusion parts. In addition, I am wondering how is the performance of full cells using a-VO₂ as cathode compared with the reported systems?

Point-by-point response to the reviewers' comments

We thank the reviewers for their valuable comments, which we found very constructive and valuable. We have addressed all the comments to the largest possible extent. For ease of reference, our responses are shown in **blue**, and the resulting changes to the manuscript are in **red**.

Reviewer #3 (Remarks to the Author):

The responses to the reviewers' comments are appropriate and the revised version is acceptable for publication in my opinion.

Response: We sincerely appreciate your strong support of the revised manuscript and valuable comments.

Reviewer #4 (Remarks to the Author):

The authors have addressed all my concerned issues. However, I suggest that the authors had better polish the language before the acceptance.

Response: We thank you for your valuable comments and help to improve the manuscript. We have now worked on both language and readability by involving a native English speaker for language corrections. We sincerely hope that the language and readability have improved.

For the comments from Reviewer #1 and #2:

After reading the authors' response to reviewers, it looks that they have addressed the questions raised by the reviewers.

The reviewer 1's main concern is that "the construction of the simulation model and the determination of the amorphous phase is very challenging." And also he/she thought that "the material with strong pseudocapacitive storage behavior makes it less likely a candidate for high-capacity cathode materials for K-ion batteries." The reviewer 2 was also mainly concerned about computational aspect. According to the reviewers' valuable comments, the authors have supplemented the relevant evidences, which considerably improved the quality of this work.

However, before accepting, the authors should further polish the logics and language of this manuscript. In particular, the author should further polish the abstract, introduction and conclusion parts. In addition, I am wondering how is the performance of full cells using a-VO₂ as cathode compared with the reported systems?

Response: We are very grateful for your positive comments. As you noted, we have adequately addressed the questions raised by reviewers 1 and 2 by supplementing relevant evidence, which considerably improved the quality of this work. To further enhance the logic and language, we have double-checked our manuscript, which has been revised and polished by a native English speaker. A few examples are listed below. In addition, we have compared the amorphous VO₂ cathode's full battery performance with the reported systems.

Revision: (Abstract) Changed “Here, phase engineered VO₂ is shown as an improved potassium-ion battery cathode” to “Here, we show phase-engineered VO₂ as an improved potassium-ion battery cathode”.

(Introduction) Changed “Notwithstanding this progress, it should be mentioned that the dispersion of atoms in the nanoparticle matrix is crucial because it ultimately dictates the properties of the nanoparticle.” to “Notwithstanding this progress, the dispersion of atoms in the nanoparticle matrix ultimately dictates its properties.”

(Introduction) Changed “In addition to accommodating volumetric variations more uniformly, the amorphous configuration also renders more exposed ion channels, facilitating rapid ion intercalation and providing low internal energy and superior chemical stability.” to “In addition to accommodating more uniform volume variation, the amorphous configuration also renders more exposed ion channels, facilitates rapid ion intercalation, and provides low internal energy and superior chemical stability.”

(Introduction) Changed “The M phase VO₂ crystallizes into the monoclinic P2₁/c space group, which has a tunnel structure via VO₆ octahedra corner-sharing.” to “The M phase VO₂ crystallizes into the monoclinic P2₁/c space group with a tunnel structure via corner-sharing VO₆ octahedra.”

(Introduction) Changed “we picked VO₂ as an example cathode for potassium-ion batteries (PIBs) to perform material modification from the perspective of phase engineering.” to “we pick VO₂ as an example cathode for potassium-ion batteries (PIBs) to perform material modification from the phase engineering perspective.”

(Page 14) Changed “the pseudocapacitance contribution for the total stored charge in

VO₂ (a) increases from 83% to 94% (Supplementary Fig. 12c), and these values are obviously higher than those in VO₂ (B) (from 55% to 79%, Supplementary Fig. 12g).”

to “the pseudocapacitance contribution for the total stored charge in VO₂ (a) increases from 83% to 94% (Supplementary Fig. 12c). These values are obviously higher than those in VO₂ (B) (from 55% to 79%, Supplementary Fig. 12g).”

(Discussion) Changed “pseudocapacitance behavior plays a more important role in the capacity contribution” to “pseudocapacitance behavior plays a more prominent role in the capacity contribution.”

(Discussion) Changed “The results of this study provide new perspectives for electrode materials on rechargeable alkali batteries.” to “This study provides promising perspectives for electrode materials used in rechargeable alkali batteries.”

(Page 18) The full battery performance with VO₂ (a) as the cathode is compared with that of the reported systems (Supplementary Fig. 22a and b). The VO₂ (a) displays one of the best battery performances, exhibiting good rate capability and cycling performance.

The comparison of full battery performance of VO₂ (a) as the cathode with the reported systems is shown in Supplementary Fig. 22 (see also Fig. R1 below).

Fig. R1 Comparison of full battery a) rate capability and b) cycling performance of

VO₂ (a) as the cathode with the reported systems¹⁻⁷.

The corresponding references:

- 1 Deng, L. *et al.* Defect-free potassium manganese hexacyanoferrate cathode material for high-performance potassium-ion batteries. *Nat. Commun.* **12**, 2167 (2021).
- 2 Chong, S. *et al.* Potassium nickel iron hexacyanoferrate as ultra-long-life cathode material for potassium-ion batteries with high energy density. *ACS Nano* **14**, 9807-9818 (2020).
- 3 Deng, T. *et al.* Self-templated formation of P2-type K_{0.6}CoO₂ microspheres for high reversible potassium-ion batteries. *Nano Lett.* **18**, 1522-1529 (2018).
- 4 Choi, J. U. *et al.* K_{0.54}[Co_{0.5}Mn_{0.5}]O₂: New cathode with high power capability for potassium-ion batteries. *Nano Energy* **61**, 284-294 (2019).
- 5 Lin, X., Huang, J., Tan, H., Huang, J. & Zhang, B. K₃V₂(PO₄)₂F₃ as a robust cathode for potassium-ion batteries. *Energy Storage Mater.* **16**, 97-101 (2019).
- 6 Zhu, Y.-H. *et al.* Reconstructed orthorhombic V₂O₅ polyhedra for fast ion diffusion in K-ion batteries. *Chem* **5**, 168-179 (2019).
- 7 Wu, L. *et al.* Layered superconductor Cu_{0.11}TiSe₂ as a high-stable K-cathode. *Adv. Funct. Mater.*, 2109893 (2021).